# A Random Matrix Theory of Masked Self-Supervised Learning

Arie Wortsman [1]   Federica Gerace [2]   Bruno Loureiro [1]   Yue M. Lu [3]

## Abstract

In the era of transformer models, masked self-supervised learning (SSL) has become a foundational training paradigm. A defining feature of masked SSL is that training aggregates predictions across many masking patterns, giving rise to a joint, matrix-valued predictor rather than a single vector-valued estimator. This object encodes how coordinates condition on one another and poses new analytical challenges. We develop a precise high-dimensional analysis of masked modeling objectives in the proportional regime where the number of samples scales with the ambient dimension. Our results provide explicit expressions for the generalization error and characterize the spectral structure of the learned predictor, revealing how masked modeling extracts structure from data. For spiked covariance models, we show that the joint predictor undergoes a Baik–Ben Arous–Péché (BBP)-type phase transition, identifying when masked SSL begins to recover latent signals. Finally, we identify structured regimes in which masked self-supervised learning provably outperforms PCA, highlighting potential advantages of SSL objectives over classical unsupervised methods.

## 1. Introduction

Self-supervised learning (SSL) — a training paradigm in which models learn useful representations from unlabeled data by exploiting the data itself as a source of supervision — has emerged as a foundational component of the recent success of transformer architectures. By avoiding the need for manual annotations, SSL retains many of the benefits traditionally associated with supervised learning while avoiding reliance on labeled data. Consequently, SSL is widely adopted as a pretraining paradigm for learning general-purpose representations that substantially accelerate the optimization of downstream tasks, especially in data-scarce settings.

A canonical example of a self-supervised learning task is masked language modeling (MLM), in which a neural network is trained to predict masked tokens in text using the remaining tokens as contextual information (Devlin et al., 2019a; Howard & Ruder, 2018; Radford et al., 2018; Brown et al., 2020; OpenAI, 2024). For example, given the sentence "*The capital of France is Paris*", a typical MLM task would be to teach the model to infer that we are speaking about the capital of a country from the context "France" and "Paris" from the masked sentence "*The [MASK] of France is Paris*". Masked language modeling is the core training framework of LLMs in the BERT family (Devlin et al., 2019b; Liu et al., 2019), where it is used to train state-of-the-art NLP encoders for which bi-directional context is crucial, as well as in vision transformers (Bao et al., 2021; He et al., 2022).

Recent works (Rende et al., 2024b;a) have numerically shown that, when trained with MLM, transformers learn the conditional distribution of a missing token given the observed ones by sequentially inferring positional and semantic relationships across the entire text. However, several fundamental questions remain open. For instance: *How much data is required to achieve good generalization performance in MLM? How does this performance depend on the underlying structure of the data, and in particular on the correlations among tokens?*

In this work, we investigate masked self-supervised learning in its simplest declination: learning real-valued sequence data with a matrix-valued linear predictor. More precisely, given $n$ real-valued sequences $x_1, \ldots, x_n \in \mathbb{R}^d$, where $d$ corresponds to the length of the sequence, we train a family of ridge-regularized linear predictors $X \mapsto X\hat{A}$ under the constraint that no coordinate of the sequence can be used to predict itself (this will be made precise in the next section). In this context, **our main contributions** are:

- **Asymptotic limit for training and generalization errors:** We derive a sharp asymptotic characterization of

[1]Departement d'Informatique, École Normale Supérieure, PSL & CNRS [2]Department of Mathematics, University of Bologna, Piazza di Porta San Donato 5, 40126, Bologna (BO), Italy [3]Applied Mathematics, Harvard John A. Paulson School of Engineering and Applied Sciences. Correspondence to: Arie Wortsman <arie.wortsman@psl.ens.eu>.

*Proceedings of the 43rd International Conference on Machine Learning*, Seoul, South Korea. PMLR 306, 2026. Copyright 2026 by the author(s).

the training and generalization performance of the aggregate predictor $\hat{A}$ in the high-dimensional regime where $n, d \to \infty$ with a fixed ratio $n/d \to \alpha > 0$. The resulting asymptotic limit depends only on the population covariance of the data, yielding a precise and interpretable description of which structural properties of the data enable successful generalization in masked self-supervised regression (SSR).

- **Deterministic equivalent for the predictor:** Beyond performance metrics, we establish a high-dimensional deterministic equivalent for (the resolvent of) the matrix-valued aggregate predictor $\hat{A}$ itself, which allows us to characterize how self-supervised learning encodes and exploits the underlying data geometry. Technically, this result relies on the analysis of a random matrix defined as an aggregate of a diverging number of correlated predictors. This matrix does not belong to standard random-matrix ensembles, and our analysis may be of independent interest.

- **Spiked vs AR(1):** We further provide an in-depth analysis of two prototypical statistical models. In a **spiked covariance model**, we show that principal component analysis (PCA) strictly outperforms masked self-supervised learning, and we establish a BBP-type transition in the asymptotic spectrum of $\hat{A}$ occurring at the same threshold as the corresponding transition for the sample covariance matrix. In sharp contrast, for a **first-order autoregressive model**, we prove that masked self-supervised regression can strictly dominate PCA in performance if the number of directions in PCA is not close to the dimension. We further provide a detailed analysis of the effective inductive bias induced by strong temporal correlations, elucidating the mechanisms by which self-supervision leverages sequential structure.

Together, our results elucidate the mechanisms by which masked self-supervised learning exploits correlations in the training data, and provide a principled comparison between SSR and classical spectral approaches, revealing their respective strengths and limitations in the high-dimensional regime.

**Further Related Works:** Deterministic equivalents for sample covariance matrices have a long history in the random matrix theory literature, starting with the seminal work of (Marčenko & Pastur, 1967) and followed by several generalizations to the anisotropic settings (Bai & Zhou, 2008; Burda et al., 2004; Knowles & Yin, 2017; Rubio & Mestre, 2011; Louart & Couillet, 2018; Chouard, 2022).

In the context of statistics and machine learning theory, RMT techniques have been used to derive insight into a manifold of high-dimensional models, including linear regression (Dobriban & Wager, 2018; Wu & Xu, 2020; Hastie et al., 2022; Cheng & Montanari, 2024), kernels (Hu et al.,

2024; Xiao et al., 2022; Misiakiewicz & Saeed, 2024; Lu & Yau, 2025), random features (Pennington & Worah, 2017; Louart et al., 2018; Liao & Couillet, 2018; Mei et al., 2022; d'Ascoli et al., 2020; Fan & Wang, 2020; Benigni & Péché, 2021; Schröder et al., 2023; Schröder et al., 2024; Defilippis et al., 2024; Atanasov et al., 2024; Latourelle-Vigeant & Paquette, 2026), one-step analysis of feature learning (Ba et al., 2022; Moniri et al., 2023; Dandi et al., 2025) and in-context learning (Lu et al., 2025). In the unsupervised context, RMT has been used to study mainly PCA (Baik et al., 2005; Reiss & Wahl, 2020) (Reiss & Wahl, 2020) and clustering methods (Couillet & Liao, 2022). More recently, RMT techniques were used to investigate sequential and time series data in (Ilbert et al., 2024), and a matrix-valued least-squares problem in (Fan & Ma, 2024).

**Notation:** Given a matrix $X \in \mathbb{R}^{n \times d}$, we consistently index by $i, j \in [n]$ the rows of $X$, and by $k, l \in [d]$ its columns, e.g. $X = (x_{ik})_{\substack{1 \le i \le n \\ 1 \le k \le d}}$. We denote by $X_i \in \mathbb{R}^d$ the $i$th row of $X$ and $X_k \in \mathbb{R}^n$ the $k$th column of $X$. In the same way, for $i \in [n]$, we denote by $X^{(-i)} \in \mathbb{R}^{(n-1) \times d}$ the matrix obtained by removing the $i$th row, and by $X^{(-k)} \in \mathbb{R}^{n \times (d-1)}$ the matrix obtained by removing the $k$th column.

## 2. Setting

Let $x_1, \ldots, x_n \in \mathbb{R}^d$ denote $n$ data points in $\mathbb{R}^d$, for $n, d \in \mathbb{N}$, and let $X = [x_1, \ldots, x_n]^\top \in \mathbb{R}^{n \times d}$ denote the data matrix. Here, each $x_i$ can be thought as a sequence of (real valued) tokens or pixels of length $d$, in the case of text or image data, for instance.

As motivated in the introduction, we are interested in the problem of masked self-supervised learning: given a sequence $x \in \mathbb{R}^d$ and an arbitrary coordinate $k \in [d]$ (the "*mask*"), the goal is to predict the masked coordinate $x_k$ from the remaining $[d] \setminus \{k\}$ coordinates. Here, we focus on the simplest instance of this problem where we perform self-supervised ridge regression with regularization $\lambda > 0$:

$$\hat{a}_k = \underset{\substack{a \in \mathbb{R}^d \\ a_k = 0}}{\arg \min} \frac{1}{n} ||X_k - Xa||_2^2 + \lambda ||a||_2^2, \quad (1)$$

where the constraint $a_k = 0$ makes sure that we do not use the $k$th coordinate to predict itself.

*Remark* 2.1. The problem in Equation (1) corresponds exactly to the regularized maximum likelihood problem for the case in which the sequence $x = (x_1, \ldots, x_d)$ is jointly Gaussian and the variance is fixed.

Equation (1) is a strictly convex problem for $\lambda > 0$ — denote by $\hat{a}_k$ its minimizer for each $k \in [d]$. We can define the following aggregate prediction matrix:

$$\hat{A} := [\hat{a}_1, \hat{a}_2, \ldots, \hat{a}_d] \in \mathbb{R}^{d \times d}. \quad (2)$$

Note that by construction $\text{diag}(\hat{A}) = 0$. This matrix, which we will refer to as the *self-supervised ridge matrix* (SSR matrix), contains aggregated information about the correlations in the training data $X$.

*Remark* 2.2 (Relationship with factored attention). As discussed in (Rende et al., 2024b), for this real-valued sequence task the matrix-valued predictor $\hat{A} \in \mathbb{R}^{d \times d}$ can be interpreted as the attention map of a single layer of *factored self-attention*, with an untrained value matrix fixed to the identity. Despite its simplicity, factored attention has been empirically shown to outperform standard attention mechanisms in some contexts, including protein contact prediction (Bhattacharya et al., 2020) and the approximation of ground states of many-body quantum systems (Viteritti et al., 2023a; Rende et al., 2023; Viteritti et al., 2023b).

In the following, we will be mainly interested in two questions: (i) what properties of the data are captured by the SSR matrix $\hat{A}$? (ii) How well does $\hat{A}$ perform reconstruction of the masked entries at a fixed training data budget, as quantified by the total reconstruction population risk:

$$L(\hat{A}) = \frac{1}{d}\mathbb{E}_x \left[ \|x - \hat{A}^T x\|^2 \right], \qquad (3)$$

where the expectation is taken over a new sequence $x \in \mathbb{R}^d$ that is not in the training set.

*Remark* 2.3. The total reconstruction error is also commonly used as the generalization error in other unsupervised learning tasks, in particular for PCA [see, e.g (Reiss & Wahl, 2020) and (El Hanchi et al., 2025)].

Complementary, we also define the total reconstruction empirical risk:

$$\hat{L}_n(\hat{A}) = \frac{1}{nd}\|X - XA\|_F^2, \qquad (4)$$

where $X \in \mathbb{R}^{n \times d}$ is the training data matrix in Equation (1). We will refer to Equation (3) and Equation (4) as the generalization and training errors, respectively.

Our main results provide sharp answers to both questions (i) and (ii) in the *high-dimensional limit*, where the number of samples $n$ and the sequence length $d$ grow to infinity proportionally, with $n/d \to \alpha = \Theta_d(1)$ denoting the sample complexity. More precisely, we will derive a *deterministic equivalent* for the resolvent of the SSR matrix $\hat{A}$ and asymptotic limits of the corresponding risks $L(\hat{A})$, $\hat{L}_n(\hat{A})$. These results allow us to precisely characterize how the aggregated predictor encodes the geometry of the underlying data distribution, and how this geometry governs the total reconstruction error.

## 3. Main Results

Our first result concerns the structural properties of the aggregated predictor matrix $\hat{A}$ defined in Equation (2). While

vector-valued ridge estimators of the form Equation (1) have been extensively analyzed in the random matrix theory literature, the aggregate estimator studied here exhibits fundamentally different behavior. In particular, although each coordinate-wise minimizer $\hat{a}_k$ solves a standard ridge regression problem, all such estimators are trained on the same data matrix $X$, inducing strong statistical dependencies across coordinates. As a consequence, existing RMT results that treat each $\hat{a}_k$ in isolation are insufficient to characterize the joint behavior of the matrix-valued estimator $\hat{A}$, its spectrum, and the associated reconstruction risk. A global analysis that explicitly accounts for these correlations is therefore required. To that end, we first derive a joint expression for the minimizer $\hat{A}$:

**Lemma 3.1.** *For $\lambda > 0$, let $\hat{\Sigma} = \frac{1}{n}X^\top X \in \mathbb{R}^{d \times d}$ denote the sample-covariance matrix and let*

$$Q(\lambda) = (\hat{\Sigma} + \lambda I_d)^{-1}$$

*denote the resolvent of $\hat{\Sigma}$. Then*

$$\hat{A} = I_d - Q(\lambda)\Lambda, \qquad \Lambda = [\text{diag}(Q(\lambda))]^{-1}.$$

We refer the reader to Appendix A for the proof.

*Remark* 3.2. Note that the matrix $\hat{A}$ is, in general, not symmetric as $Q(\lambda)$ and $\Lambda$ do not commute in general. However, by applying the similarity transformation $S \to \Lambda^{\frac{1}{2}} S \Lambda^{-\frac{1}{2}}$ to the matrix $Q(\lambda)\Lambda$, we obtain the matrix $\Lambda^{\frac{1}{2}} Q(\lambda) \Lambda^{\frac{1}{2}}$, which is a symmetric matrix. Then $\hat{A}$ has the same spectrum as $I_d - \Lambda^{\frac{1}{2}} Q(\lambda) \Lambda^{\frac{1}{2}}$, and therefore has a real-valued spectrum.

Lemma 3.1 shows that $\hat{A}$ depends on the data only through the resolvent of the empirical covariance matrix. This representation naturally suggests an approach based on random matrix theory. The main technical challenge, however, is that this dependence appears through the product $Q(\lambda)\Lambda$, where $\Lambda$ is a diagonal matrix defined as a nonlinear function of the resolvent. Although $\Lambda$ admits a deterministic equivalent in the high-dimensional limit, its finite-dimensional dependence on $Q(\lambda)$ precludes a direct application of standard random matrix results. Our analysis therefore proceeds by first establishing a sharp concentration of $\Lambda$ around its deterministic limit, and then leveraging a linearization argument to control the resulting structured product and characterize the behavior of the aggregated predictor.

### 3.1. The Generalization and Training Errors

With the description of $\hat{A}$ in Lemma 3.1, we can characterize the generalization properties of the SSR predictor. A first natural statistical question is whether this is a consistent estimator, i.e. whether the estimator $\hat{A}$ reaches vanishing generalization error in the classical statistical limit where $n \to \infty$ at fixed $d = \Theta_n(1)$. The answer is negative: To

see this, recall that given $k \in [d]$, we can always decompose $x_k = \mathbb{E}[x_k|x^{(-k)}] + (x_k - \mathbb{E}[x_k|x^{(-k)}])$, and $\mathbb{E}[x_k|x^{(-k)}]$ is the best estimator of $x_k$ given the rest of the coordinates. Then, if $x_k - \mathbb{E}[x_k|x^{(-k)}]$ is non-zero, there will always be a part that is not predictable by the rest of the coordinates. As an example, if the data were sampled from a Gaussian distribution with a full-rank covariance, then $x_k - \mathbb{E}[x_k|x^{(-k)}]$ is always non-zero.

**Lemma 3.3** (Approximation error). *Assume* $\Sigma := \mathbb{E}[xx^\top] \succ 0$. *Then, the approximation error is given by:*

$$L_{\text{App}} := \min_{\substack{A \in \mathbb{R}^{d \times d} \\ \text{diag}(A)=0}} L(A) = \frac{1}{d}\text{Tr}([\text{diag}(\Sigma^{-1})]^{-1}) \quad (5)$$

*and it is attained by* $A_{\text{App}} := I_d - \Sigma^{-1}[\text{diag}(\Sigma^{-1})]^{-1}$.

*Remark* 3.4. The approximation error defined above is the best achievable error in the SSR hypothesis: a linear predictor in the form of a square matrix with zero-diagonal.

Lemma 3.3 shows that in general the lowest achievable error is non-zero. Moreover, note that in the particular case where $\Sigma$ is a diagonal matrix, the best possible estimator is $\hat{A} \equiv 0$. This is intuitive, as when coordinates are not correlated there is no information on a coordinate in the remaining. This highlights that structure is fundamental for successful SSR, and raises the question of what properties of the data geometry are particularly important. To get an intuition for this question, we can take inspiration from the case of Gaussian data, where we have that for $k \in [d]$, $\text{Var}(x_k|x_{-k}) = 1/(\Sigma^{-1})_{kk}$. Hence, the matrix $[\text{diag}(\Sigma^{-1})]^{-1}$ can be interpreted as a measure of how predictable (or explainable) each coordinate is with respect to the rest.

Now that we understand what the best achievable error is, we move to addressing the performance of SSR at finite sample complexity. Our first main result is to show that the risk achieved by the SSR predictor $L(\hat{A})$ admits a fully deterministic characterization in the proportional high-dimensional limit where $d \to \infty$ with $n = \Theta(d)$, depending only on the population covariance $\Sigma$, the asymptotic ratio $n/d \to \alpha$ and the regularization strength $\lambda > 0$.

**Theorem 3.5** (Asymptotic Limit of the risk). *Let* $X = Z\Sigma^{\frac{1}{2}} \in \mathbb{R}^{n \times d}$, *with* $\Sigma \succeq 0$, *and* $\|\Sigma\|_{\text{op}}$ *bounded, and* $Z$ *having independent entries, with mean* 0, *variance* 1, *and* $4 + \varepsilon$*-th bounded moments. Let* $\text{df}_1^\Sigma := \text{Tr}(\Sigma(\Sigma + \kappa I_d)^{-1})$ *and* $\text{df}_2^\Sigma := \text{Tr}(\Sigma^2(\Sigma + \kappa I_d)^{-2})$ *denote the degrees of freedom. Let* $\kappa_\star(\lambda)$ *denote the (unique) solution of the self-consistent equation:*

$$\kappa = \lambda + \frac{\kappa}{n}\text{df}_1^\Sigma(\kappa). \quad (6)$$

*Then, as* $n, d \to \infty$ *at fixed* $n/d \to \alpha = \Theta_d(1)$, *the normal-ized generalization error converges in probability to:*

$$L(\hat{A}) \to L_1\left(1 + \frac{\text{df}_2^\Sigma(\kappa_\star(\lambda))}{n - \text{df}_2^\Sigma(\kappa_\star(\lambda))}\right)$$

*where*

$$L_1 = \frac{1}{d}\text{Tr}\left[\bar{D}^2(\Sigma + \kappa_\star(\lambda)I_d)^{-1}\Sigma(\Sigma + \kappa_\star(\lambda)I_d)^{-1}\right],$$

*and*

$$\bar{D} = \bar{D}(\Sigma, \kappa_\star(\lambda)) := [\text{diag}((\Sigma + \kappa_\star(\lambda)I_d)^{-1})]^{-1}.$$

*Remark* 3.6. It is important to highlight that computing this asymptotic limit from existing limits in the literature for the minimizer $\hat{a}_k, k \in [d]$ (e.g., (Dobriban & Wager, 2018; Wu & Xu, 2020; Hastie et al., 2022)) would require summing a diverging number of individual formulas, one for each $k \in [d]$. Instead, Theorem 3.5 gives us a closed formula for the loss associated with the aggregate predictor.

To get some intuition for the formulas in Theorem 3.5, it is instructive to consider the ridge-less limit $\lambda \to 0^+$ when $\alpha > 1$. In this case, $\kappa(\lambda) \to 0$, so $L_1 \to L_{\text{App}}$ defined in Lemma 3.3, which is independent of $n$. Then:

$$L(\hat{A}) - L_1 \sim L_1\frac{\text{df}_2(0)}{n - \text{df}_2(0)} = L_{\text{App}}\frac{1}{n/d - 1}. \quad (7)$$

From the above, we can see an explicit dependence of the generalization error on $\alpha$. More generally, we can study how $L(\hat{A})$ converges to the approximation error by computing the degrees of freedom, which depend only on the eigenvalues of $\Sigma$. Then, the general performance of the SSR estimator will be determined by $L_{\text{App}}$, which is a highly structure-dependent quantity (as described above), and the speed at which the convergence will happen depends on the eigenvalues of $\Sigma$.

Away from this particular limit, we can numerically verify Theorem 3.5 against finite-size simulations. The top Figure 1 shows the generalization error for self-supervised ridge regression for Gaussian data with a structured covariance $\Sigma$, which corresponds to a linear auto-regressive process. We plot both the empirical generalization error and the asymptotic limit of Theorem 3.5, showing a remarkable agreement even for modest sizes $d = 200$. The dotted lines below represent the approximation error given by Lemma 3.3. Note that the behavior of the generalization error varies depending on the value of $\rho$, changing both behavior of the curves and its asymptotic value. This model will be discussed in depth in Section 4.2. Finally, a similar characterization as the one in Theorem 3.5 holds for the training error defined in Equation (4).

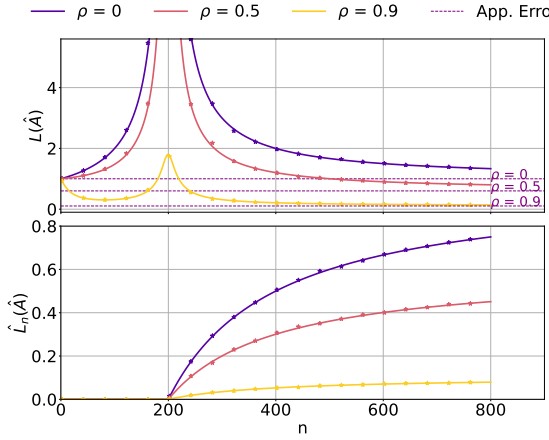

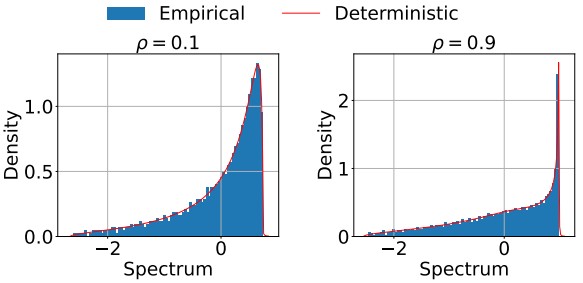

Figure 2. Spectrum of the SSR matrix $\hat{A}$ for Gaussian Data with a Toeplitz Covariance parametrized by $\rho \in (0,1)$, that is: $\Sigma_{i,j} = \rho^{|i-j|}$. The blue lines are the empirical spectrum, while the red lines are predicted by Theorem 3.8. In both experiments $\lambda = 0.01$, $d = 1000$ and $\alpha = 3$.

Figure 1. Generalization and Training Error for the SSR estimator for Gaussian data with a Toeplitz covariance $\Sigma_{i,j} = \rho^{|i-j|}$, for different values of $\rho$. The value $\rho = 0$ corresponds to $\Sigma = I_d$. Solid lines correspond to the asymptotic limit, while dots correspond to the empirical error. In all curves, $d = 200$ and $\lambda = 10^{-4}$.

**Theorem 3.7.** *Under the same assumptions and notation as Theorem 3.5, as $n, d$ grow to $\infty$ proportionally as $\alpha = \frac{n}{d}$, the normalized training error converges in probability to*

$$\hat{L}(\hat{A}) \to \frac{\kappa(\lambda)}{\lambda d}\bar{L}_1 - \frac{\kappa(\lambda)^2}{\lambda d}\bar{L}_2 - \frac{\kappa(\lambda)^2}{\lambda d(n - \mathrm{df}_2^\Sigma(\kappa(\lambda)))}\bar{L}_3,$$

*where, denoting $\bar{Q}(\kappa) = (\Sigma + \kappa I_d)^{-1}$, we have:*

$$\bar{L}_1 = \mathrm{Tr}\left(\bar{D}^2\bar{Q}(\kappa)\right); \quad \bar{L}_2 = \mathrm{Tr}\left(\bar{D}^2\bar{Q}(\kappa)^2\right),$$

*and*

$$\bar{L}_3 = \mathrm{Tr}\left(\bar{D}^2\bar{Q}(\kappa)^2\Sigma\right)\mathrm{Tr}\left(\bar{Q}(\kappa)^2\Sigma\right).$$

Theorem 3.7 is illustrated at the bottom of Figure 1. For all values of $\rho$, the asymptotic limit in Theorem 3.7 shows excellent agreement with the finite-size training error. From this figure, it is clear that the origin of the double descent peak in Figure 1 coincides with the point in which the estimator interpolates the training data, akin to the standard double descent phenomena in supervised learning (Mei et al., 2022; Gerace et al., 2020; Hastie et al., 2022).

### 3.2. Asymptotic spectrum of the SSR matrix

Beyond performance characterization, a natural question is how the structure of the data distribution is reflected in the SSR estimator $\hat{A}$. As shown in (Rende et al., 2024b), for simple data models the attention matrix learned by single-layer architectures trained with masked self-supervised objectives is closely related to the inverse of the data covariance. This behavior is closely aligned with the structure that emerges in our self-supervised ridge regression setting.

As noted in Remark 3.2, we can study the eigenvalues of $\hat{A}$ by studying the eigenvalues of the matrix $\Lambda^{\frac{1}{2}}Q(\lambda)\Lambda^{\frac{1}{2}}$, for $\Lambda = [\mathrm{diag}(Q(\lambda))]^{-1}$. Our second main result is a characterization of the asymptotic spectrum of $\Lambda^{\frac{1}{2}}Q(\lambda)\Lambda^{\frac{1}{2}}$, and therefore that of the SSR predictor $\hat{A}$.

**Theorem 3.8.** *For $\lambda > 0$ and $\alpha = \frac{n}{d}$, let $\tilde{m}(\lambda)$ be the unique solution of the self-consistent equation:*

$$\tilde{m}(\lambda) = \left(\lambda + \frac{1}{n}\mathrm{Tr}\left(\Sigma(\tilde{m}\Sigma + I_d)^{-1}\right)\right)^{-1}. \quad (8)$$

*For $z \in \mathbb{C}^+$, let $\chi$ be the solution to the self-consistent equation*

$$\chi = \frac{1}{n}\mathrm{Tr}\left(\Sigma(-\lambda I_d - \frac{1}{1-\chi}\Sigma + \bar{D}/z)^{-1}\right). \quad (9)$$

*where $\bar{D}$ is a diagonal matrix with entries $\bar{D}_{k,k} = \frac{\lambda}{(\tilde{m}(\lambda)\Sigma + I_d)_{k,k}^{-1}}$. Then, the matrix:*

$$\mathcal{G}(z) = \left[\bar{D}^{\frac{1}{2}}\left(\frac{1}{1-\chi}\Sigma + \lambda I_d\right)^{-1}\bar{D}^{\frac{1}{2}} - zI_d\right]^{-1} \quad (10)$$

*is a deterministic equivalent for the resolvent:*

$$G(z) := (\Lambda^{\frac{1}{2}}Q(\lambda)\Lambda^{\frac{1}{2}} - zI_d)^{-1}. \quad (11)$$

*As an immediate consequence, for any $z \in \mathbb{C}^+$, the Stieltjes transform of the empirical spectral measure of $\Lambda^{\frac{1}{2}}Q(\lambda)\Lambda^{\frac{1}{2}}$ evaluated at $z$ converges in probability to $\frac{1}{d}\mathrm{Tr}\,\mathcal{G}(z)$.*

For the proof of Theorem 3.8 we refer the reader to Appendix B. Figure 2 illustrates the empirical spectral density of $\hat{A}$ with the deterministic equivalent given in Theorem 3.8 for Gaussian data generated from an AR(1) process. (See Section 4.2 for details of this model.)

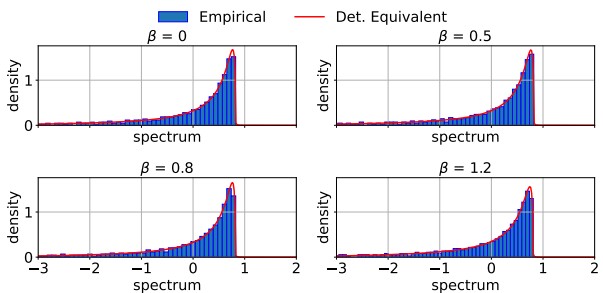

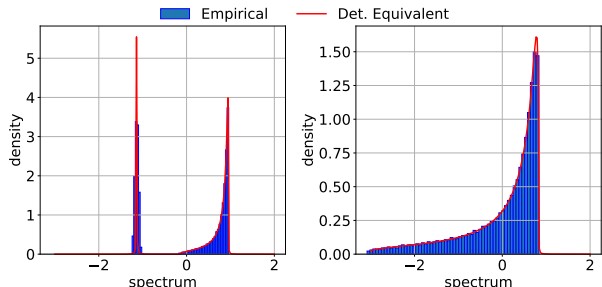

*Figure 3.* Spectral density of $\hat{A}$ and the predicted spectral density from Corollary 3.9 for Gaussian data with covariance $\Sigma = C_\beta \mathrm{diag}(1, 2^{-\beta}, \ldots, d^{-\beta})$, with $C_\beta$ such that $\mathrm{Tr}(\Sigma) = 1$. In all plots $\lambda = 10^{-5}$ and $d = 500$.

*Figure 4.* Empirical spectral density of $\hat{A}$ for Gaussian, isotropic data, compared with the spectrum predicted by Theorem 3.8. The dimension is $d = 2000$ and $\lambda = 0.01$. On the left, $\alpha = 0.6$, and on the right, $\alpha = 1.5$.

### 3.2.1. UNIVERSALITY OF THE SPECTRUM

An interesting consequence of Theorem 3.8 arises when we focus on the case where the covariance $\Sigma$ is a diagonal matrix and we consider the ridge-less limit $\lambda \to 0^+$. From Lemma 3.3, we know that this task is equivalent to simply fitting noise.

**Corollary 3.9.** *Assume $\Sigma$ is a diagonal matrix. Then, for $\alpha > 1$ and $\lambda \to 0$, the deterministic equivalent for the resolvent in Theorem 3.8 is independent of $\Sigma$:*

$$\mathcal{G}(z) = \frac{1}{(1 - \frac{1}{\alpha})(1 - \chi) - z} I_d,$$

*where $\chi$ solves:*

$$\chi^2 + (z - 1)\chi + \frac{z}{\alpha - 1} = 0. \tag{12}$$

*In particular, the spectral density of the SSR matrix $\hat{A}$ is supported in the interval $\left[\frac{-2}{\sqrt{\alpha}-1}, \frac{2}{\sqrt{\alpha}+1}\right]$.*

The fact that $\mathcal{G}(z)$ is independent of the values in $\Sigma$ means the asymptotic spectrum in the ridge-less limit is universal. Figure 3 illustrates this universality for the family of diagonal power-law covariances parametrized by an exponent $\beta > 0$. Despite the heavy-tailed spectrum of the population covariance, as $\lambda$ approaches $0^+$ the different spectra collapse to the same universal shape for $\alpha > 1$. As a sanity check for our results on the spectrum of the SSR matrix, consider the simplest possible setting of i.i.d. isotropic Gaussian data $x_i \sim \mathcal{N}(0, I_d)$. In this case, the asymptotic spectral density of $\hat{\Sigma}_n = \frac{1}{n} X^T X$ is the Marchenko-Pastur law, $\tilde{m}(\lambda)$ is the solution of

$$-\frac{\lambda}{\alpha} \tilde{m}(\lambda)^2 - (1 - \frac{1}{\alpha} + \lambda)\tilde{m}(\lambda) + 1 = 0. \tag{13}$$

Then in this case, the spectral density of the SSR matrix $\hat{A}$ is just a transformation of the Marchenko-Pastur distribution.

To illustrate this, Figure 4 shows the empirical spectral density of $\hat{A}$ with the deterministic spectral density stated in Theorem 3.8. In the case where $\alpha < 1$, there are fluctuations around the atom on the left. These fluctuations decrease as $d$ goes to infinity.

## 4. Case Studies

In the last section, we showed that the high-dimensional limit of the SSR predictor can be characterized by a deterministic equivalent depending on the data only through the properties of the population covariance $\Sigma$. We now turn to an in-depth discussion of two particular examples of structured data which are popular in the machine learning literature, corresponding to two different types of tasks: one in which the coordinates of $x$ are correlated through an underlying low-dimensional structure, and one in which $x$ follows an autoregressive structure. Since we are considering a linear SSL, a natural linear unsupervised learning benchmark is PCA. For the reader's convenience, we recall that the PCA estimator with $p$ principal components (or simply, $p$ directions) is defined as:

$$A_p^{\mathrm{PCA}} = \arg \min_{A \succ 0, \mathrm{rank}(A) = p} \frac{1}{dn} \sum_{i=1}^{n} \|x_i - Ax_i\|^2. \tag{14}$$

### 4.1. Spiked Covariance

In this section, we consider the case where $\Sigma = I_d + \theta vv^\top$, with $v \in \mathbb{S}^{d-1}$, and $\theta \geq 0$. Known as the *spiked covariance model*, it is a popular model in the statistics literature for a situation where one needs to learn a signal $v \in \mathbb{R}^d$ in otherwise noisy data (Donoho et al., 2018).

The first question we want to answer in this setting is: What is the performance of $\hat{A}$ compared to PCA in the task of reconstructing a random vector with covariance given by $\Sigma = I_d + \theta vv^\top$? How does it change depending on the signal-to-noise ratio $\theta$? We first answer this question in the classical limit $n \to \infty$.

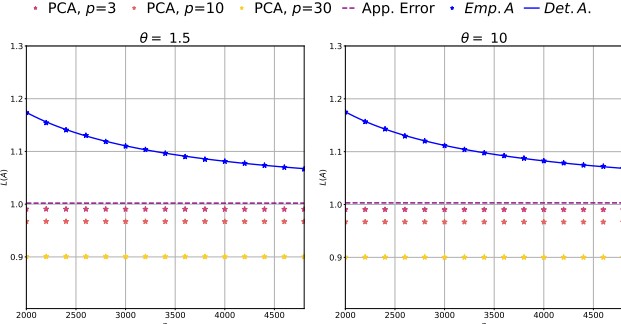

Figure 5. Comparison of the SSR estimator and PCA for a spiked covariance $\Sigma = I_d + \theta v v^\top$, for $v \sim \mathrm{Unif}(\mathbb{S}^{d-1})$. In the experiment, $d = 300$, $\lambda = 0.01$ and PCA is applied for $p \in \{3, 10, 30\}$.

**Lemma 4.1.** *Consider centered data with distribution satisfying the assumptions of Theorem 3.5, and covariance $\Sigma = I_d + \theta v v^\top$, for $\theta \geq 0$ and $v \in \mathbb{S}^{d-1}$. Let $L(\mathrm{SSR})$ denote the infinite-data generalization error in Equation (3) for Self-Supervised Ridge Regression, and $L(\mathrm{PCA}_p)$ denote the infinite-data generalization error for PCA with $p$ directions. Then, for any values of $\theta, \lambda$ and $p \geq 1$, we have:*

$$L(\mathrm{PCA}_p) < L(\mathrm{SSR}).$$

This lemma gives us insight on how strong the correlation in the data needs to be in order for SSR to achieve a reasonable performance. Recall that for isotropic data $\Sigma = I_d$ the best SSR matrix is $\hat{A} = 0$, since there is no correlation in the data. We can see the spike $\theta v v^\top$ as a perturbation of this case, which introduces correlations between the entries of $x$. Indeed, for a constant SNR $\theta = \Theta_d(1)$, the correlations are vanishing in $d$ (because $\|v\|_2 = 1$) so the correlations are rather weak in the high-dimensional limit. Lemma 4.1 illustrates this by comparing the empirical error of PCA with the asymptotic limit of the generalization error in Theorem 3.5 for different values of the SNR $\theta$ in the large $n$ limit. As it can be seen, even the approximation error in this case is not better than $L(\hat{A}) = 1$, which corresponds to no reconstruction.

We now turn to the question of whether signatures of the low-dimensional structure of $\Sigma$ can be detected in the spectrum of the SSR matrix $\hat{A}$. Recall that the sample covariance matrix $\hat{\Sigma}_n$ in this model undergoes a Baik-Ben Arous-Peché (BBP) transition (Baik et al., 2005): the spectrum of $\hat{\Sigma}$ will not identify the spiked direction $v$ unless $\theta > \theta^\star$, for some critical value $\theta^\star > 0$. In other words, under certain values of $\theta$ the spectral distribution of $\hat{\Sigma}$ is no different than if the model was sampled from pure noise (isotropic data). As $\theta$ is increased, there exists a value $\theta^\star$ after which an isolated eigenvalue pops out of the Marchenko-Pastur bulk of $\hat{\Sigma}_n$.

Since the matrix $\hat{A}$ is related to the sample covariance

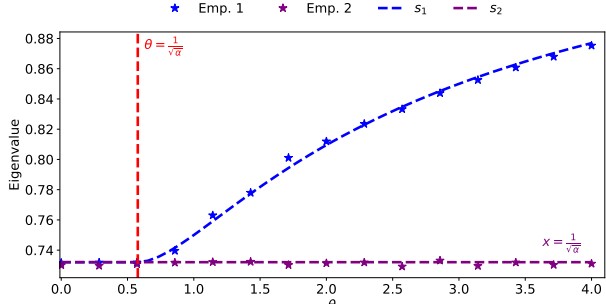

Figure 6. Baik-Ben Arous-Peché transition for the SSR matrix $\hat{A}$ for Gaussian data with covariance $\Sigma = I_d + \theta v v^T$, for $v \sim \mathrm{Unif}(\mathbb{S}^{d-1})$ and varying $\theta$. The value of $\lambda = 10^{-5}$, $d = 2000$, and $\alpha = \frac{n}{d} = 2$.

through the resolvent $Q(\lambda)$, a natural question is whether a similar BBP transition can be observed for $\hat{A}$. Building on the deterministic equivalents found in 3.8 and Corollary 3.9, we prove the following result concerning this phase transition.

**Proposition 4.2.** *Let $s_1, s_2$ denote the top two eigenvalues of the SSR matrix $\hat{A}$ in the ridgeless limit, and let $\xi$ be the solution of Equation (12). Then*

$$s_1 \xrightarrow{n,d \to \infty} \begin{cases} 1 - z^* & \text{if } \theta > \frac{1}{\sqrt{\alpha}} \\ \frac{2}{1+\sqrt{\alpha}} & \text{otherwise ,} \end{cases} \quad (15)$$

*where $z^*$ solves equation:*

$$\frac{z\alpha}{(\alpha - 1)(1 - \chi)} = \frac{1}{1 + \theta}, \quad (16)$$

*for $\chi$ solving Equation (89), and $s_2 \xrightarrow{n,d \to \infty} \frac{2}{1+\sqrt{\alpha}}$.*

*Remark* 4.3. The classical BBP transition for the covariance also occurs at $\theta^* = \sqrt{\frac{d}{n}} = \sqrt{\frac{1}{\alpha}}$, so the phase transition occurs at the same SNR in both models. The second eigenvalue $s_2$ converges to the top of the bulk in the case where $\Sigma = I_d$ (see Corollary 3.9).

The result in Proposition 4.2 is illustrated in Figure 6.

### 4.2. Auto-Regressive Model of Order 1

As a second example, we now consider a case in which $x$ follows an auto-regressive structure common in sequence modeling. The simplest such example is stationary autoregressive Process of order one (a.k.a. AR(1) process):

$$x_k = \rho x_{k-1} + \varepsilon_k, \quad k \in [d], \quad (17)$$

where $\varepsilon_k$ are i.i.d centered random variables with variance $\frac{1}{1-\rho^2}$ so that $\mathrm{Var}(x_k) = 1$, and $0 < \rho < 1$ is a parameter.

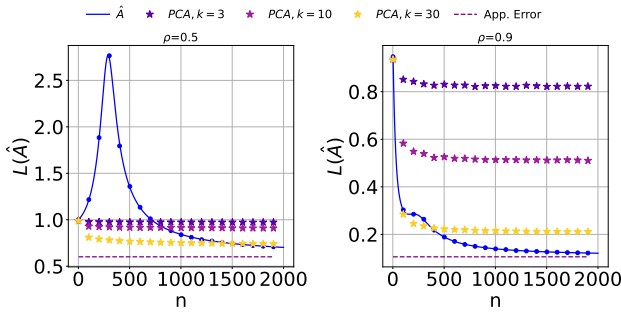

*Figure 7.* Empirical generalization error for PCA and the self-supervised ridge estimator for Gaussian data with a Toeplitz covariance, for different values of $\rho$. In both pictures, $d = 200$ and $\lambda = 0.01$.

Note that the $\rho \to 0^+$ limit corresponds to the isotropic setting.

The covariance of AR(k) processes are Toeplitz matrices (see, e.g Appendix 3 in (Potters & Bouchaud, 2020)): A matrix $\Sigma \in \mathbb{R}^{d \times d}$ such that its entry $\Sigma_{jk}$ depends on the difference $|j - k|$. In the AR(1) case:

$$\Sigma_{jk} = \mathbb{E}\left[x_j x_k\right] = \rho^{|j-k|}, \tag{18}$$

which describes the correlation between $x_j$ and $x_k$ that decays exponentially as $|j - k|$ grows. The spectrum and properties of this matrix have been studied, for example, in (Kac et al., 1953) and (Narayan & Shastry, 2020). Our first result for this setting shows that unlike the spiked covariance case, in the population limit SSR outperforms PCA unless the number of directions $p = \Theta(d)$. This is summarized in the following proposition.

**Proposition 4.4.** *Consider centered data with distribution satisfying the assumptions of Theorem 3.5, and covariance of the form in Equation (18), with a fixed parameter $\rho \in (0, 1)$. Let $L(\text{SSR})$ and $L(\text{PCA}_p)$ denote the population limit of the generalization error for self-supervised ridge regression and PCA with $p$ directions, respectively. Let $\gamma := \frac{p}{d}$. Then, as $d$ grows to infinity and $\lambda \to 0^+$, we have*

$$L(\text{PCA}_p) < L(\text{SSR}), \tag{19}$$

*if and only if*

$$\frac{2}{\pi} \arctan\left(\frac{1-\rho}{1+\rho} \tan\left(\frac{\pi}{2}\left(\frac{(2\rho)^2}{(1+\rho)^2 + (1-\rho)^2}\right)\right)\right) \leq \gamma.$$

Note that the LHS in the inequality in Proposition 4.4 is increasing in $\rho$. Therefore, as the SNR $\rho$ in the AR(1) model in Equation (17) becomes bigger, it becomes harder for PCA to outperform the self-supervised estimator (as measured by training sample-complexity). Moreover, in the population limit, the number of directions has to be extensive in $d$ in

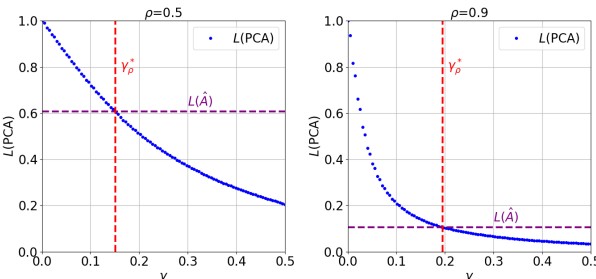

*Figure 8.* Generalization error for PCA and the self-supervised Ridge estimator for Gaussian data with a Toeplitz covariance, for different values of $\rho$. The sample size is fixed at $n = 20.000$, the dimension is $d = 300$ and $\lambda = 10^{-5}$. For PCA with $p$ directions, $\gamma = \frac{p}{d}$.

order to reach the same performance the SSR predictor. For an illustration of the curve $\gamma^\star(\rho)$, we refer the reader to Figure 9 in the Appendix. This result is intuitive in the sense that, since corelation between coordinates decay with their distance, each coordinate becomes more predictable by using its neighbors.

Proposition 4.4 is also illustrated in Figure 7 and Figure 8. Figure 7 compares these two estimators for increasing values of $\rho$ in Equation (17), corresponding to an increasing dependence of $x_k$ on $x_{k-1}$. As the sample complexity $\alpha = n/d$ grows, we see that for PCA to be at the same generalization error as the self-supervised estimator, a greater number of directions $p$ is needed. Its also possible to see that for higher $\rho$ the peak of $L(\hat{A})$ is also lower. This is explained by the fact that, as computed in Appendix E, for high-dimensional Toeplitz matrices, the degrees of freedom are approximately:

$$\text{df}_2^\Sigma(\kappa) \approx \frac{\sqrt{1-\rho^2}}{4\sqrt{\kappa}}. \tag{20}$$

In both panels of Figure 8, $\gamma_\rho^\star$ denotes the quantity on the LHS of Proposition 4.4. For both values of $\rho$, $L(\text{SSR}) < L(\text{PCA})$, approximately until $\gamma$ reaches $\gamma_\rho^*$.

## 5. Conclusion

In this work, we studied masked self supervised models in their simplest declination: self-supervised ridge regression. We derived asymptotic limits for the generalization error and the asymptotic spectral distribution of the matrix-valued SSR estimator. With this characterization, we proved the importance of structure in the data for this type of tasks, and tested this on two classical statistical models: the spiked covariance model and an auto-regressive process of order 1. The limitations of our model come from the fact that we are studying linear regression. Therefore, a potential future direction is the study of a genuinely non-linear self-supervised model.

## Acknowledgements

We would like to thank Lenka Zdeborová for insightful discussions. This research was motivated by discussions during the "Huddle on Learning and Inference from Structured Data" at ICTP Trieste in 2023. We would like to thank the huddle organizers Jean Barbier, Manuel Sáenz, Subhabrata Sen, and Pragya Sur for their hospitality and many helpful discussion. BL and AW were supported by the French government, managed by the National Research Agency (ANR), under the France 2030 program with the project references "ANR-23-IACL-0008" (PR[AI]RIE-PSAI) and "ANR-25-CE23-5660" (MAPLE), as well as the Choose France - CNRS AI Rising Talents program. AW was also funded by the PSL Graduate Program in Computer Science. The work of YML is supported by a Harvard College Professorship, by the Harvard FAS Dean's Fund for Promising Scholarship, and by DARPA under grant DIAL-FP-038. FG is supported by European Union-NextGenerationEU (NGEU) and she is partially supported by project SERICS (PE00000014) under the MUR National Recovery and Resilience Plan.

## Impact Statement

This paper presents work whose goal is to advance our theoretical understanding of Machine Learning. There are many potential societal consequences of our work, none which we feel must be specifically highlighted here.

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

# A. Preliminaries

Recall the definition of $\hat{A}$:

$$\hat{A} := [\hat{a}_1, \ldots, \hat{a}_d] \in \mathbb{R}^{d \times d}, \tag{21}$$

where $\hat{a}_k$ solves Equation (1). We can first re-write as matrix optimization problem:

$$\hat{A} = \underset{\substack{A \in \mathbb{R}^{d \times d} \\ \mathrm{diag}(A)=0}}{\arg\min} \frac{1}{n} \|X - XA^\top\|_F^2 + \lambda \|A\|_F^2. \tag{22}$$

## A.1. Proof of 3.1

We solve this problem by imposing the KKT conditions in Equation (22). Let $g_l(A) = A_{l,l}$ and $F(A) = \frac{1}{n} \|X - XA^T\|_2^2 + \lambda \|A\|_F^2$. Then, the Lagrangian is given by:

$$\mathcal{L}(A) = F(A) - \sum_{l=1}^{d} \lambda_l g_l(A) = F(A) - \mathrm{Tr}(\Lambda A) \tag{23}$$

$$= \frac{1}{n} \mathrm{Tr}\left((I_d - A^T)X^T X(I_d - A)\right) + \lambda \mathrm{Tr}(A^T A) - \mathrm{Tr}(\Lambda A) \tag{24}$$

where $\Lambda = \mathrm{diag}(\lambda_1, \ldots, \lambda_d)$. Then, differentiating with respect to $A$, we have:

$$\frac{\partial}{\partial A} \mathcal{L}(A) = -\frac{2}{n} X^T X + \frac{2}{n} X^T X A + 2\lambda A + \Lambda, \tag{25}$$

and imposing the stationarity condition of the Lagrangian we get:

$$\frac{\partial \mathcal{L}}{\partial A} = -\frac{2}{n} X^T X + \frac{2}{n}(X^T X + \lambda I_d)A + \Lambda = 0. \tag{26}$$

Denote $Q(\lambda) := (\hat{\Sigma} + \lambda I_d)^{-1}$. Then, solving for $A$ gives:

$$A = (\frac{1}{n} X^T X + \lambda I_d)^{-1}(\frac{1}{n} X^T X + \frac{1}{2}\Lambda) = Q(\lambda)(\hat{\Sigma} + \frac{1}{2}\Lambda). \tag{27}$$

In order to obtain $\Lambda$, we impose the restriction of the optimization problem:

$$\mathrm{diag}(\hat{A}) = \mathrm{diag}\left((\hat{\Sigma} + \lambda I_d))^{-1}(\hat{\Sigma} + \frac{1}{2}\Lambda)\right) = 0 \tag{28}$$

$$\iff \lambda_k Q(\lambda)_{k,k} = -2Q(\lambda)_{k,:}^T \hat{\Sigma}_{k,:} \tag{29}$$

$$\iff \lambda_k = \frac{-2}{Q(\lambda)_{k,k}} Q(\lambda)_{k,:}^T \hat{\Sigma}_{k,:}, \tag{30}$$

where we used the fact that $\Lambda$ is diagonal. Then, by defining $D(\lambda) \in \mathbb{R}^{d \times d}$ the diagonal matrix with entries:

$$D(\lambda)_{k,k} = \frac{\mathrm{diag}(Q(\lambda)\hat{\Sigma})_{k,k}}{Q(\lambda)_{k,k}}, \tag{31}$$

we can write:

$$\hat{A} = Q(\lambda)(\hat{\Sigma} - D(\lambda)). \tag{32}$$

By adding and subtracting the $\frac{\lambda}{n}$ in (32), we also get:

$$\hat{A} = I_d - Q(\lambda)(D(\lambda) + \lambda I_d). \tag{33}$$

Note that

$$D(\lambda) + \lambda I_d = \frac{\text{diag}(Q(\lambda)\hat{\Sigma})}{\text{diag}(Q(\lambda))} + \lambda I_d \tag{34}$$

$$= \frac{1}{\text{diag}(Q(\lambda))}\left(\text{diag}(Q(\lambda)\hat{\Sigma}) + \lambda \text{diag}(Q(\lambda))\right) \tag{35}$$

$$= \frac{1}{\text{diag}(Q(\lambda))}. \tag{36}$$

Hence, we finally write:

$$\hat{A} = I_d - Q(\lambda)[\text{diag}(Q(\lambda))]^{-1}. \tag{37}$$

## A.2. Concentration

**Lemma A.1.** *Consider a matrix $X \in \mathbb{R}^{n \times d}$, where each row is an independent sample from $\mathcal{N}(0, \Sigma)$, and $\hat{A}$ the SSR matrix. As $n, d \to \infty$, with high probability, we have:*

$$\|\hat{A} - (I_d - Q(\lambda)\bar{D}\|_{\text{op}} \overset{d \to \infty}{\to} 0,$$

*where $\bar{D} \in \mathbb{R}^{d \times d}$ is a diagonal matrix with entries:*

$$\bar{D}_{k,k} := \frac{\lambda}{[(I_d + \tilde{m}(\lambda)\Sigma)^{-1}]_{k,k}},$$

*where $\tilde{m}$ solves the self-consistent equation*

$$\tilde{m} = \left(\lambda + \frac{1}{n}\text{Tr}\left(\Sigma(\tilde{m}(\lambda)\Sigma + I_d)^{-1}\right)\right)^{-1}. \tag{38}$$

*Proof.* Let $X = Z\Sigma^{\frac{1}{2}} \in \mathbb{R}^{n \times d}$, where $Z$ is a matrix with standard gaussian, independent entries, and let $Q(\lambda) := (\hat{\Sigma} + \lambda I_d)^{-1}$, for $\lambda > 0$. By Lemma 3.1, we have:

$$\hat{A} = I_d - Q(\lambda)[\text{diag}(Q(\lambda))]^{-1}. \tag{39}$$

We will to prove that, for $d$ big enough, we can replace $[\text{diag}(Q(\lambda))]^{-1}$ by $[\text{diag}(\bar{Q}(\lambda))]^{-1}$, where

$$\bar{Q}(\lambda) := -\frac{1}{\lambda}(\tilde{m}(\lambda)\Sigma + I_d)^{-1}, \tag{40}$$

is the deterministic equivalent of $\bar{Q}(\lambda)$ (Theorem 2.6, (Couillet & Liao, 2022)). Let $\bar{A} := \hat{A} = I_d - Q(\lambda)[\text{diag}(\bar{Q}(\lambda))]^{-1}$. We have:

$$\|\hat{A} - \bar{A}\|_{\text{op}} = \|Q(\lambda)([\text{diag}(Q(\lambda))]^{-1} - [\text{diag}(\bar{Q}(\lambda))]^{-1})\|_{\text{op}}. \tag{41}$$

By using the fact that

$$\|Q(\lambda)\|_{\text{op}} \leq \frac{1}{\lambda}, \tag{42}$$

we get:

$$\|\hat{A} - \bar{A}\|_{\text{op}} \leq \frac{1}{\lambda} \max_{k \in [d]} \left|\frac{1}{Q(\lambda)_{k,k}} - \frac{1}{\bar{Q}(\lambda)_{k,k}}\right|. \tag{43}$$

Now using the identity $A^{-1} - B^{-1} = A^{-1}(B - A)B^{-1}$:

$$\|\hat{A} - \bar{A}\|_{\text{op}} \leq C \max_{k \in [d]} \left|Q(\lambda)_{k,k} - \bar{Q}(\lambda)_{k,k}\right| \tag{44}$$

$$\leq C \max_{k \in [d]} \left|e_k^T Q(\lambda) e_k - e_k^T \bar{Q}(\lambda) e_k\right|, \tag{45}$$

where we used that $\|\hat{\Sigma}\|_{\text{op}} \leq C$ with high probability ((Vershynin, 2010), Theorem 5.44). From here, we conclude by using the fact that $\bar{Q}(\lambda)$ is the deterministic equivalent of $Q(\lambda)$, so $a^T Q(\lambda) b \to a^T \bar{Q}(\lambda) b$, for $a, b \in \mathbb{S}^{d-1}$ and a union-bound on each $k \in [d]$. $\square$

# B. The Spectrum of $\hat{A}$

With the results in Appendix A, we we can proceed to study the spectrum of the SSR matrix $\hat{A}$, given by:

$$\hat{A} := I_d - Q(\lambda)\Lambda, \quad \Lambda = [\text{diag}(Q(\lambda))]^{-1} \tag{46}$$

The first challenge we note is the second term: We have the product of $Q(\lambda)$ with a diagonal matrix $\Lambda$, which are both random but not independent. We will overcome this complication by leveraging the fact that the diagonal of the resolvent is, under mild conditions, very concentrated. This is where we will first apply Lemma A.1. With this, we will be able to replace the diagonal matrix $[\text{diag}(Q(\lambda))]^{-1}$ by a deterministic matrix $\bar{D}$, and study the matrix $I_d - Q(\lambda)\bar{D}$ instead of $\hat{A}$.

## B.1. The non-isotropic case - Proof of Theorem 3.8

In what follows, we directly study the spectrum of $\hat{A}$. Since the diagonal entries of $Q$ are nonnegative, we only need to study the spectrum of the matrix

$$\bar{D}^{\frac{1}{2}}Q(\lambda)\bar{D}^{\frac{1}{2}}, \tag{47}$$

where, by Lemma A.1

$$\bar{D}_{k,k} := \frac{\lambda}{[(I_d + \tilde{m}(\lambda)\Sigma)^{-1}]_{k,k}} \tag{48}$$

To make notation simpler, denote $\tilde{D} = \bar{D}^{\frac{1}{2}}$. For any $z \in \mathbb{C}^+$, we aim to obtain a deterministic equivalent of the resolvent

$$G(z) = \left(\tilde{D}Q(\lambda)\tilde{D} - zI_d\right)^{-1}. \tag{49}$$

To that end, we use the linearization trick. Note that:

$$\left(\tilde{D}Q(\lambda)\tilde{D} - zI_d\right)G(z) = I_d. \tag{50}$$

Now define $G_1 = Q(\lambda)\tilde{D}G(z)$, so $\tilde{D}G(z) + (-\hat{\Sigma} + \lambda I_d)G_1 = 0$. We get:

$$\tilde{D}G_1 - zG(z) = I_d. \tag{51}$$

Putting these equations together, we get the following linear system of equations:

$$\begin{pmatrix} -zI_d & \tilde{D} \\ \tilde{D} & -\hat{\Sigma} - \lambda I_d \end{pmatrix} \begin{pmatrix} G(z) \\ G_1(z) \end{pmatrix} = \begin{pmatrix} I_d \\ 0 \end{pmatrix}. \tag{52}$$

Let $H$ be the block matrix:

$$H = \begin{bmatrix} -zI & \tilde{D} \\ \tilde{D} & -\hat{\Sigma} - \lambda I \end{bmatrix}. \tag{53}$$

Then:

$$G(z) = \begin{pmatrix} I_d & 0 \end{pmatrix} \begin{pmatrix} G(z) \\ G_1 \end{pmatrix} = \begin{pmatrix} I_d & 0 \end{pmatrix} H^{-1} \begin{pmatrix} I_d \\ 0 \end{pmatrix} = [H^{-1}]_{1,1}. \tag{54}$$

We can also compute $H^{-1}$ via matrix block-inversion:

$$H^{-1} = \begin{bmatrix} G(z) & G(z)\tilde{D}Q \\ Q\tilde{D}G(z) & (\frac{\tilde{D}^2}{z} - \hat{\Sigma} - \lambda I)^{-1} \end{bmatrix} \tag{55}$$

$$= \begin{bmatrix} G(z) & G(z)\tilde{D}Q \\ Q\tilde{D}G(z) & -Q + Q\tilde{D}G(z)\tilde{D}Q \end{bmatrix}. \tag{56}$$

Note that

$$H = \underbrace{\begin{pmatrix} -zI_d & \tilde{D} \\ \tilde{D} & -\lambda I_d \end{pmatrix}}_{H_0:=} + \underbrace{\begin{pmatrix} 0 & 0 \\ 0 & -\hat{\Sigma} \end{pmatrix}}_{\text{random part}}. \tag{57}$$

The idea is to get an approximation of $\mathbb{E}[H^{-1}]$ and use it as a deterministic equivalent for $H$. For this, we first use the fact that $HH^{-1} = I_{2d}$:

$$H^{-1}H = I_{2d} \iff H^{-1}H_0 + H \begin{pmatrix} 0 & 0 \\ 0 & -\hat{\Sigma} \end{pmatrix} = I_{2d}. \tag{58}$$

Now, since $\hat{\Sigma} = \frac{1}{n}\sum_{i=1} x_i x_i^T$, we can define $\tilde{x}_i = (0_d, x_i)$, where $0_d$ denotes the $d$-zeros vector, and we will get:

$$\begin{pmatrix} 0 & 0 \\ 0 & -\hat{\Sigma} \end{pmatrix} = -\frac{1}{n}\sum_{i=1} \tilde{x}_i \tilde{x}_i^T. \tag{59}$$

Then Equation (58) becomes:

$$H^{-1}H_0 - \frac{1}{n}\sum_{i=1}^{n} H^{-1}\tilde{x}_i \tilde{x}_i^T = I_{2d}, \tag{60}$$

and taking expectation, since $H_0$ is deterministic:

$$\mathbb{E}[H^{-1}]H_0 - \frac{1}{n}\sum_{i=1}^{n} \mathbb{E}\left[H^{-1}\tilde{x}_i \tilde{x}_i^T\right] = I_{2d} \tag{61}$$

Now, we define the leave-one-out version of $H$. For each $i \in [n]$, let

$$H_{\backslash i} = H_0 - \frac{1}{n}\sum_{j \neq i} \tilde{x}_j \tilde{x}_j^T. \tag{62}$$

Then, by the Sherman-Morrison formula we have that:

$$H^{-1} = H_{\backslash i}^{-1} - \frac{H_{\backslash i}^{-1}\tilde{x}_i \tilde{x}_i^T H_{\backslash i}^{-1}}{1 - \frac{1}{n}\tilde{x}^T H_{\backslash i}^{-1} \tilde{x}_i}. \tag{63}$$

Then, we can write the second term in Equation (61) (without the expectation) as:

$$H^{-1}\tilde{x}_i \tilde{x}_i^T = H_{\backslash i}^{-1}\tilde{x}_i \tilde{x}_i^T + \frac{H_{\backslash i}^{-1}\tilde{x}_i \tilde{x}_i^T H_{\backslash i}^{-1}}{1 - \frac{1}{n}\tilde{x}^T H_{\backslash i}^{-1} \tilde{x}_i}\tilde{x}_i \tilde{x}_i^T \tag{64}$$

$$= H_{\backslash i}^{-1}\tilde{x}_i \tilde{x}_i^T \frac{1}{1 - \frac{1}{n}\tilde{x}_i^T H_{\backslash i}^{-1} \tilde{x}_i}. \tag{65}$$

Note that by concentration of quadratic forms ((Couillet & Liao, 2022), Lemma 2.1), with very high probability:

$$\frac{1}{n}\tilde{x}_i^T H_{\backslash i}^{-1} \tilde{x}_i = \frac{1}{n}\mathrm{Tr}\left(\Sigma [H_{\backslash i}^{-1}]_{2,2}\right) + o_d(1) \tag{66}$$

$$= \frac{1}{n}\mathrm{Tr}\left(\Sigma \mathbb{E}[H_{\backslash i}^{-1}]_{2,2}\right) + o_d(1), \tag{67}$$

where the last line follows from the fact that $\mathbb{E}[H_{\backslash i}^{-1}]$ is a deterministic equivalent for $H_{\backslash i}^{-1}$, and $[\cdot]_{2,2}$ denotes the second block of the matirx (coordinates $d+1$ to $2d$). Let $\chi = \frac{1}{n}\mathrm{Tr}\left(\Sigma \mathbb{E}[H_{\backslash i}^{-1}]\right)$. Then applying Equation (65) in Equation (61):

$$\mathbb{E}[H^{-1}]H_0 - \frac{1}{n(1-\chi)}\sum_{i=1}^{n} \mathbb{E}\left[H_{\backslash i}^{-1}\tilde{x}_i \tilde{x}_i^T\right] = I_{2d} \tag{68}$$

$$\iff \mathbb{E}[H^{-1}]H_0 - \frac{1}{n(1-\chi)}\sum_{i=1}^{n} \mathbb{E}\left[H_{\backslash i}^{-1}\right] \begin{pmatrix} 0 & 0 \\ 0 & \Sigma \end{pmatrix} = I_{2d}, \tag{69}$$

$$\iff \mathbb{E}[H^{-1}]H_0 + \mathbb{E}\left[H_{\backslash i}^{-1}\right] \begin{pmatrix} 0 & 0 \\ 0 & \frac{1}{(1-\chi)}\Sigma \end{pmatrix} = I_{2d}. \tag{70}$$

where in the second to last line we used the independence of $H_{\backslash i}$ and $\tilde{x}_i$. Now, using that $\mathbb{E}[H_{\backslash i}^{-1}] = \mathbb{E}[H^{-1}] + o_n(1)$, we get:

$$\mathbb{E}[H^{-1}] \left( H_0 + \begin{pmatrix} 0 & 0 \\ 0 & -\dfrac{1}{(1-\chi)}\Sigma \end{pmatrix} \right) = I_{2d} + o_n(1). \tag{71}$$

Therefore:

$$\mathbb{E}[H^{-1}] \sim \begin{pmatrix} -zI & \tilde{D} \\ \tilde{D} & -\dfrac{1}{(1-\chi)}\Sigma - \lambda I_d \end{pmatrix}^{-1}. \tag{72}$$

At last, recalling Equation (67), by the matrix inversion lemma we have:

$$\mathbb{E}[H_{\backslash i}^{-1}]_{2,2} \sim \left[ \begin{pmatrix} -zI & \tilde{D} \\ \tilde{D} & -\dfrac{1}{(1-\chi)}\Sigma - \lambda I_d \end{pmatrix}^{-1} \right]_{2,2} = \left( \dfrac{\tilde{D}^2}{z} - \dfrac{1}{1-\chi}\Sigma - \lambda I_d \right)^{-1}, \tag{73}$$

and then, asymptotically $\chi$ solves::

$$\chi = \dfrac{1}{n}\mathrm{Tr}\left( \Sigma \left( \dfrac{\tilde{D}^2}{z} - \dfrac{1}{1-\chi}\Sigma - \lambda I_d \right)^{-1} \right). \tag{74}$$

In conclusion, we've find that:

$$H^{-1} \sim \begin{pmatrix} -zI & \tilde{D} \\ \tilde{D} & -\dfrac{1}{(1-\chi)}\Sigma - \lambda I_d \end{pmatrix}^{-1}, \tag{75}$$

and then, since $G(z)$ is the upper left block of $H^{-1}$, we have:

$$G(z) \sim \left( \tilde{D} \left( \dfrac{1}{(1-\chi)}\Sigma + \lambda I_d \right)^{-1} \tilde{D} - zI_d \right)^{-1}, \tag{76}$$

where $\chi$ solves:

$$\chi = \dfrac{1}{n}\mathrm{Tr}\left( \Sigma \left( \dfrac{\tilde{D}^2}{z} - \dfrac{1}{1-\chi}\Sigma - \lambda I_d \right)^{-1} \right). \tag{77}$$

We conclude Theorem 3.8 by recalling that $\tilde{D} = \bar{D}^{\frac{1}{2}}$.

### B.2. Diagonal Population Covariance in the Ridgeless Limit - Proof of Corollary 3.9

In what follows, we work out a special case corresponding to $\Sigma$ being a diagonal matrix and the ridge parameter $\lambda \to 0^+$.

First, under assumptions on the moments and the data distribution being centered, the deterministic equivalent of the resolvent $Q(\lambda) = (\hat{\Sigma} + \lambda I_d)^{-1}$ is given by ((Couillet & Liao, 2022), Theorem 2.6):

$$\bar{Q}(\lambda) = \dfrac{(\tilde{m}(\lambda)\Sigma + I_d)^{-1}}{\lambda} = (\lambda \tilde{m}(\lambda)\Sigma + \lambda I_d)^{-1}, \tag{78}$$

where $\tilde{m}(\lambda)$ solves:

$$\tilde{m}(\lambda) = \left( \lambda + \dfrac{1}{n}\mathrm{Tr}\left( \Sigma(\tilde{m}(\lambda)\Sigma + I_d)^{-1} \right) \right)^{-1}. \tag{79}$$

Let $\dfrac{1}{1+\nu} := \lambda \tilde{m}(\lambda)$. Then:

$$\bar{Q}(\lambda) = (\dfrac{1}{\nu+1}\Sigma + \lambda I_d)^{-1}, \tag{80}$$

and $\nu$ satisfies:

$$\nu = \frac{1}{n}\mathrm{Tr}\left(\Sigma\left(\frac{1}{\nu+1}\Sigma + \lambda I_d\right)^{-1}\right). \tag{81}$$

Taking $\lambda > 0$, when $\alpha > 1$ we get:

$$\nu = \frac{1}{n}\mathrm{Tr}\left(\Sigma\left(\frac{1}{\nu+1}\Sigma\right)^{-1}\right) = \frac{d(\nu+1)}{n} = \frac{\nu+1}{\alpha}, \tag{82}$$

so the solution is $\nu = \frac{1}{\alpha-1}$. Then, in this case the matrix $\Lambda$ concentrates into:

$$\Lambda = [\mathrm{diag}\,(Q(\lambda))]^{-1} \sim \bar{D} = \frac{1}{\nu+1}\mathrm{diag}(\Sigma^{-1})^{-1}, \tag{83}$$

and since $\Sigma$ is diagonal, we conclude:

$$\bar{D} = \frac{\alpha-1}{\alpha}\Sigma. \tag{84}$$

Recall that the deterministic equivalent of Theorem Theorem 3.8 is for $\bar{D}^{\frac{1}{2}}Q(\lambda)\bar{D}^{\frac{1}{2}}$, and in this case:

$$\bar{D}^{\frac{1}{2}} = \sqrt{\frac{\alpha-1}{\alpha}}\Sigma^{\frac{1}{2}}. \tag{85}$$

Then, by Theorem 3.8, for $z \in \mathbb{C}$, the resolvent has the following deterministic equivalent:

$$G(z) \sim \left(\tilde{D}\left(\frac{1}{(1-\chi)}\Sigma\right)^{-1}\tilde{D} - zI_d\right)^{-1} \tag{86}$$

$$= \left(\frac{\alpha-1}{\alpha}\Sigma^{\frac{1}{2}}\left(\frac{1}{(1-\chi)}\Sigma\right)^{-1}\Sigma^{\frac{1}{2}} - zI_d\right)^{-1} \tag{87}$$

$$= \left(\frac{(\alpha-1)(1-\chi)}{\alpha}I_d - zI_d\right)^{-1}, \tag{88}$$

where $\chi$ solves:

$$\chi^2 + (z-1)\chi + \frac{z}{\alpha-1} = 0. \tag{89}$$

We conclude the independence of the spectrum from the elements of $\Sigma$ by noting that Equation (88) and Equation (89) are independent of $\Sigma$.

## C. Asymptotic Limits for the Training and Generalization Errors

### C.1. Proof of Lemma 3.3

In this section, we prove Lemma 3.3. Let $\Sigma \in \mathbb{R}^{d\times d}$, with $\Sigma \succ 0$. Then we can write: Indeed, we can write:

$$L(A) = \mathbb{E}_x\left[(x_i - Ax_i)^T(x_i - Ax_i)\right] \tag{90}$$

$$= \mathrm{Tr}(\Sigma) - 2\mathrm{Tr}(A\Sigma) + \mathrm{Tr}(A^T A\Sigma) \tag{91}$$

Then, the Lagrangian of this problem is:

$$\mathcal{L}(A) = \mathrm{Tr}(\Sigma) - 2\mathrm{Tr}(A^T\Sigma) + \mathrm{Tr}(A^T A\Sigma) - \mathrm{Tr}(\Lambda A), \tag{92}$$

where $\Lambda$ is a diagonal matrix. Hence, by imposing the stationarity of the Lagrangian, and using the assumption that $\Sigma$ is invertible, we get:

$$-2\Sigma + 2\Sigma A - \Lambda = 0 \implies A = \Sigma^{-1}\left(\frac{1}{2}\Lambda + \Sigma\right). \tag{93}$$

Imposing the condition $\operatorname{diag}(A) = 0$, we get:

$$A_{\text{App}} = \Sigma^{-1}(\Sigma - \operatorname{diag}(\Sigma^{-1})) = I_d - \Sigma^{-1}[\operatorname{diag}(\Sigma^{-1})]^{-1}. \tag{94}$$

By replacing $A_{\text{App}}$ in the generalization error, we get that the Approximation error is then given by:

$$L(A_{\text{App}}) = \operatorname{Tr}[\Sigma^{-1}[\operatorname{diag}(\Sigma^{-1})]^{-2}].$$

## C.2. Proof of Theorem 3.5

In this section, we prove the asymptotic limit for $L(\hat{A})$. From Equation (3) and Lemma 3.1, we have:

$$L(\hat{A}) = \frac{1}{d}\operatorname{Tr}\left((I_d - \hat{A}^T)\Sigma(I_d + \hat{A})\right) \tag{95}$$

$$= \frac{1}{d}\operatorname{Tr}\left(\Lambda Q(\lambda)\Sigma Q(\lambda)\Lambda\right), \tag{96}$$

where we recall that $Q(\lambda) = (\hat{\Sigma} + \lambda I_d)^{-1}$ and $\Lambda = [\operatorname{diag}(Q(\lambda))]^{-1}$. As we saw in Appendix A, we have that

$$\hat{A} \approx I_d - Q(\lambda)\bar{D}, \tag{97}$$

where $\bar{D} = \lambda[\operatorname{diag}(Q(\kappa(\lambda)))]^{-1}$, and $\tilde{m}(\lambda)$ is a solution to the self-consistent equation:

$$\tilde{m}(\lambda) = \left(\lambda + \frac{1}{n}\operatorname{Tr}\left(\Sigma(\tilde{m}(\lambda)\Sigma + I_d)^{-1}\right)\right)^{-1}. \tag{98}$$

Our first step in proving an asymptotic limit for $L(\hat{A})$ is to prove that we can approximate this quantity by $L(I_d - Q(\lambda)\bar{D})$. This is what we do in the following lemma.

**Lemma C.1.** *Let $X = Z\Sigma \in \mathbb{R}^{n \times d}$, with $\|\Sigma\|_{\text{op}}$ bounded, and $Z$ having independent entries, with mean $0$, variance $1$, and $4 + \varepsilon$ bounded moments. As $d$ grows to infinity, the normalized risk converges to:*

$$L(\hat{A}) \to \frac{1}{d}\operatorname{Tr}[\bar{D}Q(\lambda)\Sigma Q(\lambda)\bar{D}]. \tag{99}$$

*Proof.* Recall that $\hat{A} = I_d - Q(\lambda)\Lambda$, and as we just saw:

$$L(\hat{A}) = \operatorname{Tr}[DQ(\lambda)\Sigma Q(\lambda)D] = \operatorname{Tr}[D^2 Q(\lambda)\Sigma Q(\lambda)]. \tag{100}$$

Then:

$$L(\hat{A}) = \operatorname{Tr}[\bar{D}^2 Q(\lambda)\Sigma Q(\lambda)] + \operatorname{Tr}[\Delta^2 Q(\lambda)\Sigma Q(\lambda)], \tag{101}$$

where $\Delta = \Lambda - \bar{D}$. Therefore, we need to show that with high probability, $\operatorname{Tr}[\Delta^2 Q(\lambda)\Sigma Q(\lambda)] \to 0$, as $d \to \infty$. We have:

$$\frac{1}{d}\operatorname{Tr}[\Delta^2 Q(\lambda)\Sigma Q(\lambda)] \leq \frac{1}{d}d\|\Delta\|_{\text{op}}^2\|Q(\lambda)\Sigma Q(\lambda)\|_{\text{op}} \tag{102}$$

$$\leq C\|\Delta\|_{\text{op}}^2, \tag{103}$$

where we used that $\|\Sigma\|_{\text{op}}$ is bounded by assumption, $\|Q(\lambda)\|_{\text{op}} \leq \frac{1}{\lambda}$, and $\|\Delta\|_{\text{op}} \to 0$, as $d \to \infty$ by Lemma A.1. $\square$

Before proving the asymptotic limit, we need one more Lemma.

**Lemma C.2** (Proposition 1 in (Bach, 2024)). *Let $X = Z\Sigma \in \mathbb{R}^{n \times d}$, with $\|\Sigma\|_{\text{op}}$ bounded, and $Z$ having independent entries, with mean $0$, variance $1$. Assume that $\Sigma$ has a limiting spectral density and that $A, B$ have bounded operator norm. Then:*

$$\operatorname{Tr}\left(A(\hat{\Sigma} + \lambda I_d)^{-1}\right) \sim \frac{\kappa(\lambda)}{\lambda}\operatorname{Tr}\left(AQ(\kappa(\lambda))\right)$$

*and*

$$\text{Tr}\left(A(\hat{\Sigma}+\lambda I_d)^{-1}B(\hat{\Sigma}+\lambda I_d)^{-1}\right) \sim \frac{\kappa(\lambda)^2}{\lambda^2}\left(\text{Tr}\left(A(\Sigma+\kappa(\lambda)I_d)^{-1}B(\Sigma+\kappa(\lambda)I_d)^{-1}\right)\right.$$
$$\left. +\frac{1}{n-\text{Tr}\left(\Sigma^2(\Sigma+\kappa(\lambda)I_d)^{-2}\right)}\text{Tr}\left(A(\Sigma+\kappa(\lambda)I_d)^{-2}\Sigma\right)\text{Tr}\left(B(\Sigma+\kappa(\lambda)I_d)^{-2}\Sigma\right)\right),$$

*where $\frac{1}{\kappa(\lambda)}$ solves the self-consistent equation:*

$$\kappa(\lambda) = \lambda + \frac{\kappa(\lambda)}{n}\text{Tr}\left(\Sigma(\Sigma+\kappa(\lambda)I_d)^{-1}\right)$$

We can now proceed with the proof of Theorem 3.5. First, by Lemma C.1, we have:

$$L(\hat{A}) \sim \frac{1}{d}\text{Tr}\left(\bar{D}^2 Q(\lambda)\Sigma Q(\lambda)\right). \tag{104}$$

Then, by applying Lemma C.2 we get:

$$L(\hat{A}) \sim \frac{\kappa(\lambda)^2}{\lambda^2}\left(\text{Tr}\left(\bar{D}^2(\Sigma+\kappa(\lambda)I_d)^{-1}\Sigma(\Sigma+\kappa(\lambda)I_d)^{-1}\right)\right. \tag{105}$$

$$\left. +\frac{1}{n-\text{Tr}\left(\Sigma^2(\Sigma+\kappa(\lambda)I_d)^{-2}\right)}\text{Tr}\left(\bar{D}^2(\Sigma+\kappa(\lambda))^{-2}\Sigma\right)\text{Tr}\left(\Sigma(\Sigma+\kappa(\lambda)I_d)^{-2}\Sigma\right)\right). \tag{106}$$

Let $L_1 := \text{Tr}\left(\bar{D}^2(\Sigma+\kappa(\lambda)I_d)^{-1}\Sigma(\Sigma+\kappa(\lambda)I_d)^{-1}\right)$. Since $\Sigma$ and $(\Sigma+\kappa(\lambda)I_d)^{-1}$ commute, we can re-write Equation (106) as:

$$L(\hat{A}) \sim \frac{\kappa(\lambda)^2}{\lambda^2}L_1\left(1+\frac{1}{n-\text{Tr}\left(\Sigma^2(\Sigma+\kappa(\lambda)I_d)^{-2}\right)}L_1\text{Tr}\left(\Sigma(\Sigma+\kappa(\lambda)I_d)^{-2}\Sigma\right)\right) \tag{107}$$

By re-writing $\bar{D} = \frac{\kappa(\lambda)}{\lambda}[\text{diag}(Q(\kappa(\lambda)))]^{-1} = \frac{\kappa(\lambda)}{\lambda}\bar{D}_2$, we get:

$$L(\hat{A}) \sim L_1\left(1+\frac{\text{df}_2^\Sigma(\kappa(\lambda))}{n-\text{df}_2^\Sigma(\kappa(\lambda))}\right). \tag{108}$$

## D. Comparison with PCA in the population limit

In this section, we consider the classical statistical limit $n \to \infty$, while $d = \Theta_n(1)$. Note that in this case, $\hat{\Sigma} \to \Sigma$, and therefore all the quantities in the population limit, and in particular the resolvent, will contain the population covariance $\Sigma$. To avoid confusion, we will denote

$$Q_{\text{App}}(\lambda) := (\Sigma+\lambda I_d)^{-1}.$$

### D.1. Population Loss for Ridge Regression for $\lambda > 0$

In this section, we assume $\alpha = \frac{n}{d} = \infty$ and we train our predictor by Ridge Regression with $\lambda > 0$. In this case, by the same proof we did in A:

$$\hat{A}_{\text{App}} = I_d - Q_{\text{App}}(\lambda)[\text{diag}(Q_{\text{App}}(\lambda))]^{-1}. \tag{109}$$

Then, the Approximation error $L(\text{SSR})$ is given by:

$$L(\text{SSR}) = \frac{1}{d}\text{Tr}\left([\text{diag}(Q_{\text{App}}(\lambda))]^{-1}Q_{\text{App}}(\lambda)\Sigma Q_{\text{App}}(\lambda)[\text{diag}(Q_{\text{App}}(\lambda))]^{-1}\right) \tag{110}$$

$$= \frac{1}{d}\text{Tr}\left([\text{diag}(Q_{\text{App}}(\lambda))]^{-2}Q_{\text{App}}(\lambda)\Sigma Q_{\text{App}}(\lambda)\right). \tag{111}$$

Noting that $Q_{\text{App}}(\lambda)$ and $\Sigma$ commute, we conclude:

$$L(\text{SSR}) = \frac{1}{d}\text{Tr}\left([\text{diag}(Q_{\text{App}}(\lambda))]^{-2}\Sigma Q_{\text{App}}(\lambda)^2\right) \tag{112}$$

By using the fact that $[(\text{diag}(Q_{\text{App}}(\lambda)))]^{-2}$ is diagonal, we can summarize this as:

$$L(\text{SSR}) = \frac{1}{d}\sum_{\ell=1}^{d}\frac{(\Sigma Q_{\text{App}}(\lambda)^2)_{\ell,\ell}}{[Q_{\text{App}}(\lambda)_{\ell,\ell}]^2} \tag{113}$$

We can also write this as:

$$L(\text{SSR}) = \frac{1}{d}\sum_{\ell=1}^{d}\frac{Q_{\text{App}}(\lambda)_{\ell,\ell} - \lambda Q_{\text{App}}(\lambda)_{\ell,\ell}^2}{[Q_{\text{App}}(\lambda)_{\ell,\ell}]^2}. \tag{114}$$

In the following, we will use both Equation (113) and Equation (114).

## D.2. Population Loss for Principal Component Analysis

Taking $n \to \infty$, the PCA estimator for $p$ component becomes:

$$P_p = \frac{1}{d}\sum_{\ell=1}^{p}u_\ell u_\ell^T, \tag{115}$$

where $u_\ell u_\ell^T$ are the $p$ eigenvectors of $\Sigma$ with bigger eigenvalues. Then, the Approximation error of PCA is:

$$L(\text{PCA}) = \frac{1}{d}\text{Tr}(P_{\geq p+1}\Sigma), \tag{116}$$

where $P_{\geq p+1}$ denotes the projection into the $d - p$ lowest eigenvectors of $\Sigma$.

Using Equation (113) and Equation (116), we can now compare the Performance of PCA and Ridge in the population limit for different structures.

## D.3. Comparison for a Spiked Covariance - Proof of Lemma 4.1

In this section we study the case where the covariance of the data is given by $\Sigma = I_d + \theta v v^T$, for $\theta \geq 0$, $\|v\|_2 = 1$. It will be useful to denote $\tau = 1 + \lambda$. This way:

$$Q_{\text{App}}(\lambda) = (I_d + \theta v v^T + \lambda I_d)^{-1} = (\tau I_d + \theta v v^T)^{-1} = \frac{1}{\tau}I_d - \frac{\theta}{\tau(\tau+\theta)}v v^T, \tag{117}$$

and for every $\ell \in [d]$, the diagonal is given by:

$$Q_{\text{App}}(\lambda)_{\ell,\ell} = \frac{1}{\tau} - \frac{\theta}{\tau(\tau+\theta)}v_\ell^2 = \frac{1}{\tau}\left(1 - \frac{\theta}{\tau+\theta}v_\ell^2\right). \tag{118}$$

In order to compute the numerator in Equation (113), we note that $\Sigma$ and $Q(\lambda)_{\text{App}}^2$ commute. In particular, $v$ will be an eigenvector with eigenvalue $\frac{1+\theta}{(\tau+\theta)^2}$, and all other eigenvector will have eigenvalues $\frac{1}{\tau^2}$. Then:

$$\Sigma Q(\lambda)_{\text{App}}^2 = \frac{1}{\tau^2}I_d + \left(\frac{1+\theta}{(\tau+\theta)^2} - \frac{1}{\tau^2}\right)v v^T. \tag{119}$$

Then, we get:

$$(\Sigma Q(\lambda)_{\text{App}}^2)_{\ell,\ell} = \frac{1}{\tau^2} + \left(\frac{1+\theta}{(\tau+\theta)^2} - \frac{1}{\tau^2}\right)v_\ell^2. \tag{120}$$

This way, for a spiked covariance $\Sigma = I_d + \theta v v^T$, Equation (113) is given by:

$$L(\text{SSR}) = \frac{1}{d} \sum_{\ell=1}^{d} \frac{\frac{1}{\tau^2} + \left(\frac{1+\theta}{(\tau+\theta)^2} - \frac{1}{\tau^2}\right) v_\ell^2}{\frac{1}{\tau^2} \left(1 - \frac{\theta}{\tau+\theta} v_\ell^2\right)^2} = \frac{1}{d} \sum_{\ell=1}^{d} \frac{1 + \tau^2 \left(\frac{1+\theta}{(\tau+\theta)^2} - \frac{1}{\tau^2}\right) v_\ell^2}{\left(1 - \frac{\theta}{\tau+\theta} v_\ell^2\right)^2} \tag{121}$$

At last, to simplify, note that:

$$\tau^2 \left(\frac{1+\theta}{(\tau+\theta)^2} - \frac{1}{\tau^2}\right) = \frac{\tau^2(1+\theta) - \tau^2 - 2\tau\theta - \theta^2}{(\tau+\theta)^2} = \frac{\theta(\tau^2 - 2\tau - \theta)}{(\tau+\theta)^2}, \tag{122}$$

so

$$L(\text{SSR}) = \sum_{\ell=1}^{d} \frac{1 + \frac{\theta(\tau^2 - 2\tau - \theta)}{(\tau+\theta)^2} v_\ell^2}{\left(1 - \frac{\theta}{\tau+\theta} v_\ell^2\right)^2} = \sum_{\ell=1}^{d} \frac{1 + a v_\ell^2}{(1 - b v_\ell^2)^2}, \tag{123}$$

where we defined $a = \frac{\theta(\tau^2 - 2\tau - \theta)}{(\tau+\theta)^2}$ and $b = \frac{\theta}{\tau+\theta}$.

On the other hand, for PCA with any number of directions $p \in [d]$,

$$L(PCA) = \frac{1}{d}(d - p) = (1 - \frac{p}{d}) \tag{124}$$

Let $\gamma := \frac{p}{d}$. Then, we have:

$$L(\text{SSR}) - L(\text{PCA}) = \left(\frac{1}{d} \sum_{\ell=1}^{d} \frac{1 + a v_\ell^2}{(1 - b v_\ell^2)^2}\right) - (1 - \frac{p}{d}) \tag{125}$$

$$= \frac{p}{d} + \frac{1}{d} \sum_{\ell=1}^{d} \left(\frac{1 + a v_\ell^2}{(1 - b v_\ell^2)^2} - 1\right). \tag{126}$$

For each $\ell \in [d]$, denote

$$T_\ell = \frac{1 + a v_\ell^2}{(1 - b v_\ell^2)^2} - 1. \tag{127}$$

Then:

$$T_\ell = \frac{1 + a v_\ell^2}{(1 - b v_\ell^2)^2} - 1 \tag{128}$$

$$= \frac{1 + a v_\ell^2 - (1 - b v_\ell^2)^2}{(1 - b v_\ell^2)^2} \tag{129}$$

$$= \frac{1 + a v_\ell^2 - 1 + 2 b v_\ell^2 - b^2 v_\ell^4}{(1 - b v_\ell^2)^2} \tag{130}$$

$$= \frac{v_\ell^2 (a + 2b - b^2 v_\ell^2)}{(1 - b v_\ell^2)^2}. \tag{131}$$

Now, by the definition of $a$ and $b$:

$$a + 2b - b^2 v_\ell^2 = \frac{\theta(\tau^2 - 2\tau - \theta)}{(\tau+\theta)^2} + 2\frac{\theta}{\tau+\theta} - \frac{\theta^2}{(\tau+\theta)^2} v_\ell^2 \tag{132}$$

$$= \frac{\theta}{(\tau+\theta)^2} \left((\tau^2 + \theta(1 - v_\ell^2))\right). \tag{133}$$

Since $\|v\|_2 = 1$, we have that $1 - v_\ell^2 \geq 0$, and hence:

$$a + 2b - b^2 v_\ell^2 \geq 0. \tag{134}$$

Then, also:

$$T_\ell = \frac{v_\ell^2(a + 2b - b^2 v_\ell^2)}{(1 - bv_\ell^2)^2} \geq 0. \tag{135}$$

In particular, since $\|v\|_2 = 1$, there exists an $\ell \in [d]$ such that $a + 2b - b^2 v_\ell^2 > 0$. Therefore:

$$L(\text{SSR}) - L(\text{PCA}) \geq \frac{p}{d} > 0. \tag{136}$$

and in particular:

$$L(\text{SSR}) - L(\text{PCA}) > 0, \tag{137}$$

and therefore PCA achieves a better generalization error than Self-Supervised Ridge Regression in the population limit.

### D.4. Comparison for a Toeplitz Matrix - Proof of Proposition 4.4

In this section we will sketch the proof of Proposition 4.4. Let $\rho \in (0, 1)$, and define the Toeplitz matrix $\Sigma \in \mathbb{R}^{d \times d}$ by

$$\Sigma_{i,j} = \rho^{|i-j|}. \tag{138}$$

We want to compute the approximation error for this model which we know by Equation (114) is given by:

$$L(\text{SSR}) = \frac{1}{d} \sum_{\ell=1}^{d} \frac{Q_{\text{App}}(\lambda)_{\ell,\ell} - \lambda Q_{\text{App}}(\lambda)_{\ell,\ell}^2}{[Q_{\text{App}}(\lambda)_{\ell,\ell}]^2}. \tag{139}$$

Note that:

$$Q_{\text{App}}(\lambda)_{\ell,\ell}^2 = -\left(\frac{d}{d\lambda} Q_{\text{App}}(\lambda)\right)_{\ell,\ell} = -Q'_{\text{App}}(\lambda)_{\ell,\ell}. \tag{140}$$

Then:

$$L(\text{SSR}) = \frac{1}{d} \sum_{\ell=1}^{d} \frac{Q_{\text{App}}(\lambda)_{\ell,\ell} + \lambda Q'_{\text{App}}(\lambda)_{\ell,\ell}}{[Q_{\text{App}}(\lambda)_{\ell,\ell}]^2}. \tag{141}$$

So, what we need to compute $L(\text{SSR})$ is to compute $Q(\lambda)_{\ell,\ell}$ for each $\ell \in [d]$. At this point, we still haven't used the fact that $\Sigma$ is a Toeplitz matrix. In particular, in high dimensions, this matrix can be approximated by the following circulant matrix ((Potters & Bouchaud, 2020), Appendix A.3)

$$\tilde{\Sigma}_{i,j} = \rho^{\min\{|i-j|,|i-j+d|,|i-j-d|\}}. \tag{142}$$

Note that this only changes the element in the border of $\Sigma$. From now on, we will work with denote $\Sigma = \tilde{\Sigma}$, knowing that asymptotically both matrices share the same spectrum (see (Potters & Bouchaud, 2020), Appendix 3).

This modified version of $\Sigma$ can be diagonalized by the Fourier transform, and the eigenvector are given by:

$$[v_k]_\ell = \exp \frac{2\pi i k\ell}{d}, \tag{143}$$

where for each $k$, note that both the real and complex parts of $v_k$ are eigenvectors. As $d$ grows, we can approximate the eigenvalues of $\Sigma$ by:

$$E(x) = \frac{1 - \rho^2}{1 + \rho^2 - 2\rho\cos(\pi x)}, 0 \leq x \leq 1. \tag{144}$$

More precisely, for $\ell \in [d]$, $\lambda_\ell \sim E(\frac{\ell}{d})$. Now, recall we want to know the value of $Q(\lambda)_{\ell,\ell}$, for $\ell \in [d]$. However, since $\Sigma$ is now a circulant matrix, we have that $Q(\lambda) = (\Sigma + \lambda I)^{-1}$ is a circulant matrix. Hence, all of it's diagonal entries are the same. This way, we can write:

$$Q(\lambda)_{\ell,\ell} = \frac{1}{d} \sum_{k=1}^{d} Q(\lambda)_{k,k} = \frac{\text{Tr}(Q(\lambda))}{d}. \tag{145}$$

Then, for large $d$:

$$Q(\lambda)_{\ell,\ell} \sim \int_0^1 \frac{1}{\lambda + E(x)} dx = m_\Sigma(-\lambda), \tag{146}$$

where $m(\cdot)$ is the Stletjes transform. Using the fact that

$$t_\Sigma(z) = -zm_\Sigma(z) - 1 \implies m_\Sigma(z) = -\frac{t_\Sigma(z)}{z} - \frac{1}{z}, \tag{147}$$

for $t_\Sigma(z)$ being the T-transform ((Potters & Bouchaud, 2020), Chapter 11) , we can use ((Potters & Bouchaud, 2020), Equation A.33) to get:

$$Q(\lambda)_{\ell,\ell} \sim \frac{1}{\lambda} - \frac{1}{\lambda} \frac{1}{\sqrt{(\lambda + E^-)(\lambda + E^+)}} \tag{148}$$

$$\tag{149}$$

where $E_+ = \frac{1+\rho}{1-\rho}$, $E_- = \frac{1-\rho}{1+\rho}$. Then:

$$Q(\lambda)_{\ell,\ell}^2 = -\frac{d}{d\lambda} Q(\lambda)_{\ell,\ell} \tag{150}$$

$$= -\frac{d}{d\lambda} \left( \frac{1}{\lambda} - \frac{1}{\lambda} \frac{1}{\sqrt{(\lambda + E^-)(\lambda + E^+)}} \right) \tag{151}$$

$$= \frac{1}{\lambda^2} - \frac{\sqrt{(\lambda + E^-)(\lambda + E^+)} + \frac{\lambda(2\lambda + E^+ + E^-)}{2\sqrt{(\lambda + E^-)(\lambda + E^+)}}}{\lambda^2(\lambda + E^-)(\lambda + E^+)} \tag{152}$$

$$= \frac{1}{\lambda^2} - \frac{1}{\lambda^2\sqrt{(\lambda + E^-)(\lambda + E^+)}} - \frac{(2\lambda + E^+ + E^-)}{2\lambda(\lambda + E^-)^{\frac{3}{2}}(\lambda + E^+)^{\frac{3}{2}}}. \tag{153}$$

Then:

$$Q(\lambda)_{\ell,\ell} - \lambda Q(\lambda)_{\ell,\ell}^2 = \frac{1}{\lambda} - \frac{1}{\lambda\sqrt{(\lambda + E^-)(\lambda + E^+)}} - \left( \frac{1}{\lambda} - \frac{1}{\lambda\sqrt{(\lambda + E^-)(\lambda + E^+)}} - \frac{(2\lambda + E^+ + E^-)}{2(\lambda + E^-)^{\frac{3}{2}}(\lambda + E^+)^{\frac{3}{2}}} \right) \tag{154}$$

$$= \frac{(2\lambda + E^+ + E^-)}{2(\lambda + E^-)^{\frac{3}{2}}(\lambda + E^+)^{\frac{3}{2}}} \tag{155}$$

and:

$$\frac{Q(\lambda)_{\ell,\ell} - \lambda Q(\lambda)_{\ell,\ell}^2}{[Q(\lambda)_{\ell,\ell}]^2} = \frac{\frac{(2\lambda + E^+ + E^-)}{2(\lambda + E^-)^{\frac{3}{2}}(\lambda + E^+)^{\frac{3}{2}}}}{\frac{1}{\lambda^2}\left( 1 - \frac{1}{\sqrt{(\lambda + E^-)(\lambda + E^+)}} \right)^2} \tag{156}$$

$$= \frac{\frac{(2\lambda + E^+ + E^-)}{2(\lambda + E^-)^{\frac{3}{2}}(\lambda + E^+)^{\frac{3}{2}}}}{\frac{1}{\lambda^2}\left( \frac{\sqrt{(\lambda + E^-)(\lambda + E^+)} - 1}{\sqrt{(\lambda + E^-)(\lambda + E^+)}} \right)^2} \tag{157}$$

$$= \frac{\lambda^2(2\lambda + E^+ + E^-)}{2(\lambda + E^-)^{\frac{1}{2}}(\lambda + E^+)^{\frac{1}{2}}\left( \sqrt{(\lambda + E^-)(\lambda + E^+)} - 1 \right)^2}. \tag{158}$$

With this, we conclude

$$L(\text{SSR}) \sim \frac{\lambda^2(2\lambda + E^+ + E^-)}{2(\lambda + E^-)^{\frac{1}{2}}(\lambda + E^+)^{\frac{1}{2}}\left( \sqrt{(\lambda + E^-)(\lambda + E^+)} - 1 \right)^2}, \tag{159}$$

where we recall $E^+ = \frac{1+\rho}{1-\rho}$ and $E^- = \frac{1-\rho}{1+\rho}$. Let

$$f(\lambda, \rho) := \frac{\lambda^2(2\lambda + E^+ + E^-)}{2(\lambda + E^-)^{\frac{1}{2}}(\lambda + E^+)^{\frac{1}{2}}\left(\sqrt{(\lambda + E^-)(\lambda + E^+)} - 1\right)^2} \tag{160}$$

so that

$$L(\text{SSR}) \sim f(\lambda, \rho). \tag{161}$$

As a sanity check, note that if we take $\rho \to 0$, we get $f(\lambda, \rho) \to 1$, which we know is the case for an isotropic covariance.

On the other hand, we can compute the approximation error for PCA for a number of direction $p$ as follows. Assume that $\frac{p}{d} := \gamma$, as $d$ grows to infinity. Then, asymptotically, the approximation error of PCA will be:

$$L(\text{PCA}) = 1 - \frac{1}{d}\sum_{j=1}^{\lfloor \gamma d \rfloor} \lambda_j \sim \int_\gamma^1 E(x)dx. \tag{162}$$

By computing this integral, we get:

$$\int_\gamma^1 E(x)dx = \int_\gamma^1 \frac{1 - \rho^2}{1 + \rho^2 - 2\rho\cos(\pi x)}dx \tag{163}$$

$$= (1 - \rho^2)\int_\gamma^1 \frac{1}{1 + \rho^2 - 2\rho\cos(\pi x)}dx \tag{164}$$

$$= \frac{(1 - \rho^2)}{(1 - \rho^2)}\left(1 - \frac{2}{\pi}\arctan\left(\frac{1+\rho}{1-\rho}\tan\frac{\pi\gamma}{2}\right)\right) \tag{165}$$

$$= 1 - \frac{2}{\pi}\arctan\left(E^+\tan\frac{\pi\gamma}{2}\right), \tag{166}$$

so

$$L(\text{PCA}) \sim \left(1 - \frac{2}{\pi}\arctan\left(E^+\tan\frac{\pi\gamma}{2}\right)\right). \tag{167}$$

With this, we conclude that, asymptotically in $d$,

$$L(\text{PCA}) < L(\text{SSR}) \iff \left(1 - \frac{2}{\pi}\arctan\left(\frac{1+\rho}{1-\rho}\tan\frac{\pi\gamma}{2}\right)\right) \leq f(\lambda, \rho). \tag{168}$$

Now, we take the limit as $\lambda \to 0$ for $f(\lambda, \rho)$. Let $F(\lambda) = \sqrt{(\lambda + E^-)(\lambda + E^+)} = \sqrt{\lambda^2 + (E^+ + E^-)\lambda + 1}$. Then:

$$f(\lambda, \rho) = \frac{\lambda^2(2\lambda + E^+ + E^-)}{2(\lambda + E^-)^{\frac{1}{2}}(\lambda + E^+)^{\frac{1}{2}}\left(\sqrt{(\lambda + E^-)(\lambda + E^+)} - 1\right)^2} \tag{169}$$

$$= \frac{\lambda^2(2\lambda + E^+ + E^-)}{2F(\lambda)(F(\lambda) - 1)^2}. \tag{170}$$

Now:

$$F(\lambda) - 1 = \frac{F(\lambda)^2 - 1}{F(\lambda) + 1}, \tag{171}$$

so:

$$f(\lambda, \rho) = \frac{\lambda^2(2\lambda + E^+ + E^-)(F(\lambda) + 1)^2}{2F(\lambda)(F(\lambda)^2 - 1)^2} = \frac{\lambda^2(2\lambda + E^+ + E^-)(F(\lambda) + 1)^2}{2F(\lambda)\lambda^2\left(\lambda + (E^+ + E^-)\right)^2} \tag{172}$$

$$= \frac{(2\lambda + E^+ + E^-)(F(\lambda) + 1)^2}{2F(\lambda)\left(\lambda + (E^+ + E^-)\right)^2}. \tag{173}$$

By noting that $F(\lambda) \to 1$ as $\lambda \to 0$, we get:

$$f(\lambda, \rho) \to \frac{2}{(E^+ + E^-)}. \tag{174}$$

Then, for any fixed $\rho \in (0, 1)$, as $\lambda \to 0$ we get:

$$L(\text{PCA}) < L(\text{SSR}) \iff \left(1 - \frac{2}{\pi} \arctan\left(\frac{1+\rho}{1-\rho} \tan \frac{\pi\gamma}{2}\right)\right) \leq \frac{2}{(E^+ + E^-)} \tag{175}$$

Recall that $E^- = 1/E^+$, so:

$$L(\text{PCA}) < L(\text{SSR}) \iff \left(1 - \frac{2}{\pi} \arctan\left(E^+ \tan \frac{\pi\gamma}{2}\right)\right) \leq \frac{2E^+}{(E^+)^2 + 1} \tag{176}$$

Now, the LHS of this inequality is decreasing in $\gamma$. This means that, for a fixed value of $E^+(\rho)$, there exists a minimum value $\gamma_{\min}(\rho)$ such that $L(\text{PCA}) < L(\text{SSR})$ for all $\gamma \geq \gamma_{\min}(\rho)$. To find this value, we have:

$$\left(1 - \frac{2}{\pi} \arctan\left(E^+ \tan \frac{\pi\gamma}{2}\right)\right) \leq \frac{2E^+}{(E^+)^2 + 1} \tag{177}$$

$$\iff \frac{\pi}{2}\left(1 - \frac{2E^+}{(E^+)^2 + 1}\right) \leq \arctan\left(E^+ \tan \frac{\pi\gamma}{2}\right) \tag{178}$$

$$\iff \frac{2}{\pi} \arctan\left(\frac{1}{E^+} \tan\left(\frac{\pi}{2}\left(1 - \frac{2E^+}{(E^+)^2 + 1}\right)\right)\right) \leq \gamma \tag{179}$$

$$\iff \frac{2}{\pi} \arctan\left(\frac{1}{E^+} \tan\left(\frac{\pi}{2}\left(\frac{(E^+ - 1)^2}{(E^+)^2 + 1}\right)\right)\right) \leq \gamma \tag{180}$$

$$\iff \frac{2}{\pi} \arctan\left(\frac{1-\rho}{1+\rho} \tan\left(\frac{\pi}{2}\left(\frac{(2\rho)^2}{(1+\rho)^2 + (1-\rho)^2}\right)\right)\right) \leq \gamma \tag{181}$$

Note that $\gamma$ increases as $\rho$ moves closer to 1. This means that, as the covariance matrix becomes more structured (or, in terms of the AR(1) model, the dependence on the past is stronger), PCA needs a growing number of directions to achieve the same generalization error as self-supervised Ridge Regression.

## E. Generalization Error for Toeplitz Matrices for finite $\alpha = \frac{n}{d}$

Recall that, by Theorem 3.5, we have the following asymptotic limit for $L(\hat{A})$:

$$L(\hat{A}) \to L_1 \left(1 + \frac{\text{df}_2^\Sigma(\kappa_\star(\lambda))}{n - \text{df}_2^\Sigma(\kappa_\star(\lambda))}\right) \tag{182}$$

where

$$L_1 = \frac{1}{d} \text{Tr}\left[\bar{D}^2 (\Sigma + \kappa_\star(\lambda) I_d)^{-1} \Sigma (\Sigma + \kappa_\star(\lambda) I_d)^{-1}\right], \tag{183}$$

$$\bar{D} = \bar{D}(\Sigma, \kappa_\star(\lambda)) := [\text{diag}((\Sigma + \kappa_\star(\lambda) I_d)^{-1}))]^{-1}, \tag{184}$$

and $\kappa_\star(\lambda)$ satisfies the self-consistent equation:

$$\kappa = \lambda + \frac{\kappa}{n} \text{Tr}\left\{\Sigma(\Sigma + \kappa I_d)^{-1}\right\}. \tag{185}$$

We will first compute the degrees of freedom for any $\kappa > 0$:

$$\text{df}_2(\kappa) = \text{Tr}\left(\Sigma^2 (\Sigma + \kappa I_d)^{-2}\right) = \sum_{\ell=1}^{d} \frac{\lambda_\ell^2}{(\lambda_\ell + \lambda)^2}, \tag{186}$$

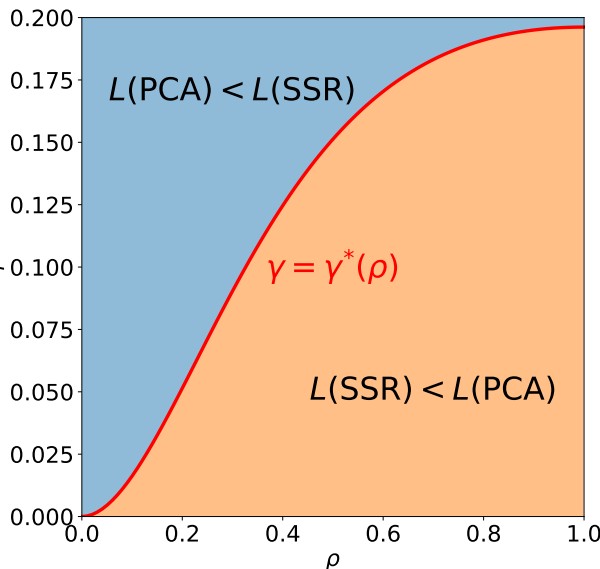

*Figure 9.* The phase transition curve defined by Equation (181). Above the curves, we have that PCA achieves a better generalization error than the SSR predictor. Below the curve, the SSR predictor achieves a better generalization error.

for $\lambda_1, \ldots, \lambda_d$ eigenvalues of $\Sigma$, and for any $\kappa > 0$. Recall that, as we saw in the last section, as $d$ grows we can approximate the eigenvalues of $\Sigma$ by:

$$E(x) = \frac{1 - \rho^2}{1 + \rho^2 - 2\rho\cos(\pi x)}, 0 \le x \le 1. \tag{187}$$

More precisely, for $\ell \in [d]$, $\lambda_\ell \sim E(\frac{\ell}{d})$. Then, we have:

$$\frac{1}{d}\mathrm{df}_2^\Sigma(\kappa) = \sum_{\ell=1}^{d} \frac{\lambda_\ell^2}{(\lambda_\ell + \lambda)^2} \tag{188}$$

$$\sim \int_0^1 \frac{E(x)^2}{(E(x) + \kappa)^2} dx. \tag{189}$$

Developing this integral we get:

$$\int_0^1 \frac{E(x)^2}{(E(x) + \kappa)^2} dx = \int_0^1 \frac{\left(\dfrac{1 - \rho^2}{1 + \rho^2 - 2\rho\cos(\pi x)}\right)^2}{\left(\dfrac{1 - \rho^2}{1 + \rho^2 - 2\rho\cos(\pi x)} + \kappa\right)^2} dx \tag{190}$$

$$= \int_0^1 \frac{\left(1 - \rho^2\right)^2}{(1 - \rho^2 + \kappa(1 + \rho^2 - 2\rho\cos(\pi x)))^2} dx \tag{191}$$

$$= (1 - \rho^2)^2 \int_0^1 \frac{1}{(1 - \rho^2 + \kappa(1 + \rho^2) - 2\kappa\rho\cos(\pi x)))^2} dx. \tag{192}$$

To compute this last integral, we recall that, by Equation A.31 in (Potters & Bouchaud, 2020):

$$\int_0^1 \frac{dx}{c - d\cos(\pi x)} = \frac{1}{\sqrt{c - d}\sqrt{c + d}}. \tag{193}$$

Then:

$$\int_0^1 \frac{dx}{(c - d\cos(\pi x))^2} = -\int_0^1 \partial_c \frac{dx}{c - d\cos(\pi x)} \tag{194}$$

$$= -\partial_c \frac{1}{\sqrt{c-d}\sqrt{c+d}} \tag{195}$$

$$= \frac{\sqrt{\frac{c+d}{c-d}} + \sqrt{\frac{c-d}{c+d}}}{(c-d)(c+d)} \tag{196}$$

$$= \frac{c}{((c-d)(c+d))^{\frac{3}{2}}}. \tag{197}$$

Then, if we go back to Equation (192):

$$\int_0^1 \frac{E(x)^2}{(E(x)+\kappa)^2} dx \sim (1-\rho^2)^2 \int_0^1 \frac{1}{(1-\rho^2 + \kappa(1+\rho^2) - 2\kappa\rho\cos(\pi x))^2} dx \tag{198}$$

$$= (1-\rho^2)^2 \frac{1-\rho^2 + \kappa(1+\rho^2)}{((1-\rho^2 + \kappa(1+\rho^2) + 2\kappa\rho)(1-\rho^2 + \kappa(1+\rho^2) - 2\kappa\rho))^{\frac{3}{2}}} \tag{199}$$

$$= (1-\rho^2)^2 \frac{1-\rho^2 + \kappa(1+\rho^2)}{((1-\rho^2 + \kappa(1+\rho)^2)(1-\rho^2 + \kappa(1-\rho)^2))^{\frac{3}{2}}}. \tag{200}$$

Then:

$$\mathrm{df}_2^\Sigma(\kappa) \sim d(1-\rho^2)^2 \frac{1-\rho^2 + \kappa(1+\rho^2)}{((1-\rho^2 + \kappa(1+\rho)^2)(1-\rho^2 + \kappa(1-\rho)^2))^{\frac{3}{2}}}. \tag{201}$$

Note that, taking $\rho \to 0$, we get $\mathrm{df}_2(\kappa) \to \frac{d}{(1+\kappa)^2}$, which corresponds to the isotropic case. Taking $\rho \to 1$, for the numerator the leading order is

$$(1-\rho^2)^2(1-\rho^2 + \kappa(1+\rho^2)) \sim 2\kappa(1-\rho^2)^2, \tag{202}$$

while for the denominator is:

$$((1-\rho^2 + \kappa(1+\rho)^2)(1-\rho^2 + \kappa(1-\rho)^2))^{\frac{3}{2}} \sim (4\kappa(1-\rho^2))^{\frac{3}{2}}. \tag{203}$$

Then, when $\rho \to 1$, by Equation (202) and Equation (203):

$$\mathrm{df}_2^\Sigma(\kappa) \sim d \frac{2\kappa(1-\rho^2)^2}{(4\kappa(1-\rho^2))^{\frac{3}{2}}} \sim d \frac{\sqrt{1-\rho^2}}{4\sqrt{\kappa}}. \tag{204}$$

Now, as $\lambda \to 0$, Equation (182) and Equation (184) become:

$$L(\hat{A}) \to L_1 \left(1 + \frac{\mathrm{df}_2^\Sigma(\kappa_\star(0))}{n - \mathrm{df}_2^\Sigma(\kappa_\star(0))}\right) \tag{205}$$

where

$$L_1 = \frac{1}{d}\mathrm{Tr}\left[\bar{D}^2(\Sigma + \kappa_\star(0)I_d)^{-1}\Sigma(\Sigma + \kappa_\star(0)I_d)^{-1}\right], \tag{206}$$

$$\bar{D} = \bar{D}(\Sigma, \kappa_\star(0)) := [\mathrm{diag}((\Sigma + \kappa_\star(0)I_d)^{-1}))]^{-1}. \tag{207}$$

For $n < d$, we have $\alpha < 1$ and $\kappa(0) > 0$, but is very close to 0 when $\alpha \to 1$. Therefore in this case, $L_1$ is almost constant when $\alpha$ is close to 1. Then, as $\alpha \to 1$

$$L(\hat{A}) \sim C(1 + \frac{1}{\frac{n}{\mathrm{df}_2^\Sigma(\kappa(0))} - 1}), \tag{208}$$

and from Equation (204):

$$L(\hat{A}) \sim C(1 + \frac{1}{\frac{4\alpha\sqrt{\kappa(0)}}{\sqrt{1-\rho^2}} - 1}). \tag{209}$$

Then, as $\rho \to 1$, the rate at which the generalization error goes to infinity when $\lambda \to 0$ changes, becoming slower.

# F. BBP Transition - Proof of Proposition 4.2

In what follows, we study the case where the population covariance takes the form of

$$\Sigma = I_d + \theta v v^\top, \tag{210}$$

where $\Sigma_0$ is the "bulk" (diagonal) matrix and $v \in \mathbb{R}^d$ is a unit-norm vector.

By the same argument we did in Appendix B.2 for the case where there is no spake, we have the following deterministic equivalent for the resolvent:

$$\bar{Q}(\lambda) = (\frac{1}{\nu+1} I_d + \lambda I_d)^{-1}, \tag{211}$$

and $\nu$ satisfies:

$$\nu = \frac{1}{n} \mathrm{Tr} \left( \left( \frac{1}{\nu+1} + \lambda I_d \right)^{-1} \right). \tag{212}$$

Taking $\lambda \to 0$, when $\alpha > 1$ we get that $\nu_{\mathrm{isotropic}} = \frac{1}{\alpha-1}$, and the diagonal matrix $\bar{D}$ becomes:

$$\bar{D}_{\mathrm{isotropic}} = \frac{1}{\nu+1} I_d. \tag{213}$$

Now, if we add a rank-one update $\theta v v^T$ to a diagonal matrix by Equation (212) we have that

$$\nu_{\mathrm{spike}} = \nu_{\mathrm{isotropic}} = \frac{1}{\alpha-1}. \tag{214}$$

Then, as $\lambda \to 0$, the deterministic equivalent for the resolvent is:

$$\bar{Q}(\lambda) = (\frac{1}{\nu+1} \Sigma + \lambda I_d)^{-1} \tag{215}$$

$$= \frac{\alpha}{\alpha-1} (I_d + \theta v v^T)^{-1} \tag{216}$$

$$= \frac{\alpha}{\alpha-1} (I_d - \frac{\theta}{1+\theta} v v^T). \tag{217}$$

Then, in this case, for each $\ell \in [d]$:

$$(\bar{D}_{\mathrm{spike}})_{\ell,\ell} = \left[ \mathrm{diag}(\bar{Q}(\lambda)) \right]_{\ell,\ell}^{-1} \tag{218}$$

$$= \frac{\alpha-1}{\alpha} \left[ \mathrm{diag}(I_d - \frac{\theta}{1+\theta} v v^T)^{-1} \right]_{\ell,\ell}^{-1} \tag{219}$$

$$= \frac{\alpha-1}{\alpha} \left( 1 - \frac{\theta}{1+\theta} v_\ell v_\ell \right)^{-1}. \tag{220}$$

*Remark* F.1. Note that if the vector $v$ is delocalized (with respect to $I_d$), then the second term is asymptotically negligible. On the other hand, if it's localized (such as $v = e_k$, for some $k \in [d]$, then this term is not negligible.

We can now examine the potential spike in our problem. Recall the matrix $\hat{A}$ can be written as:

$$\hat{A} = I - Q[\mathrm{diag}(Q)]^{-1} \tag{221}$$

As we saw before, we have that

$$\|\hat{A} - (I_d - Q(\lambda)\bar{D})\|_{\mathrm{op}} \to 0, \tag{222}$$

as $d$ grows. Therefore, for the purpose of this section we can study $I_d - Q(\lambda)\bar{D}$ instead of $\hat{A}$. The eigenvalues of this matrix are in a one-to-one correspondence with those of

$$\bar{D}^{\frac{1}{2}} Q \bar{D}^{\frac{1}{2}}. \tag{223}$$

Specifically, if $v$ is an eigenvector of $\bar{D}^{\frac{1}{2}} Q \bar{D}^{\frac{1}{2}}$ with eigenvalue $s$, then $\bar{D}^{-\frac{1}{2}} v$ is an eigenvector of $I_d - Q(\lambda)\bar{D}$ with eigenvalue $1 - s$. In what follows, we study $\bar{D}^{\frac{1}{2}} Q \bar{D}^{\frac{1}{2}}$.

We will focus on the case where the spike $v \in \mathbb{R}^d$ is generic, i.e $\|v\|_\infty = o_d(1)$. Then, in this case, the diagonal matrix $\bar{D}$ is the same as the diagonal case:

$$(\bar{D}_{\text{spike}}^{\frac{1}{2}})_{\ell,\ell} = \sqrt{\frac{\alpha-1}{\alpha}} I_d, \tag{224}$$

and the spike does not modify it.

In the isotropic case, as stated in Corollary 3.9, the bulk of the spectrum of $\bar{D}^{\frac{1}{2}}Q(\lambda)\bar{D}^{\frac{1}{2}}$ is bounded in the interval:

$$I := \left[\frac{\sqrt{\alpha}-1}{\sqrt{\alpha}+1}, \frac{\sqrt{\alpha}+1}{\sqrt{\alpha}-1}\right]. \tag{225}$$

The spike will appear in the left of $I$. By Theorem 3.8, a deterministic equivalent for the resolvent

$$G(z) := (\bar{D}^{\frac{1}{2}}Q(\lambda)\bar{D}^{\frac{1}{2}} - zI_d)^{-1}, z \in \mathbb{C}^+, \tag{226}$$

is given by:

$$\mathcal{G}(z) = \left(\bar{D}^{\frac{1}{2}}\left(\frac{1}{1-\chi}\Sigma + \lambda I_d\right)^{-1}\bar{D}^{\frac{1}{2}} - zI_d\right)^{-1}, \tag{227}$$

where $\chi$ solves:

$$\chi = \frac{1}{n}\text{Tr}\left(\Sigma(-\frac{1}{1-\chi}\Sigma + \bar{D}/z)^{-1}\right) \tag{228}$$

Replacing Equation (224) in Equation (227) we get:

$$\mathcal{G}(z) = \left(\frac{\alpha-1}{\alpha}\left(\frac{1}{1-\chi}(I_d + \theta vv^T)\right)^{-1} - zI_d\right)^{-1} \tag{229}$$

$$= \frac{\alpha}{(\alpha-1)(1-\chi)}\left((I_d + \theta vv^T)^{-1} - \frac{z\alpha}{(\alpha-1)(1-\chi)}I_d\right)^{-1} \tag{230}$$

$$= \frac{\alpha}{(\alpha-1)(1-\chi)}\left(\left(1 - \frac{z\alpha}{(\alpha-1)(1-\chi)}\right)I_d - \frac{\theta}{\theta+1}vv^T\right)^{-1} \tag{231}$$

The spike appears at $z$ such that this matrix is singular. This happens when:

$$\left(1 - \frac{z\alpha}{(\alpha-1)(1-\chi)}\right) - \frac{\theta}{\theta+1} = 0 \tag{232}$$

$$\iff \frac{z\alpha}{(\alpha-1)(1-\chi)} = 1 - \frac{\theta}{\theta+1} \tag{233}$$

$$\iff \frac{z\alpha}{(\alpha-1)(1-\chi)} = \frac{1}{\theta+1}. \tag{234}$$

Note that for $\theta = 0$, the left-limit of the bulk is $\frac{\sqrt{\alpha}-1}{\sqrt{\alpha}+1}$, and in this case, $\chi_0$ solves the quadratic equation:

$$\chi_0^2 + (z-1)\chi + \frac{z}{\alpha-1} = 0. \tag{235}$$

Note that, as $n$ and $d$ grow to infinity, the solution of Equation (228) and Equation (235) are asymptotically the same, as the spike will not change the normalized trace in the limit. Then, at $z = \frac{\sqrt{\alpha}-1}{\sqrt{\alpha}+1}$ we have:

$$\chi_0^2 + \left(\frac{\sqrt{\alpha}-1}{\sqrt{\alpha}+1} - 1\right)\chi_0 + \frac{\frac{\sqrt{\alpha}-1}{\sqrt{\alpha}+1}}{\alpha-1} = 0 \tag{236}$$

$$\iff \chi_0^2 - \frac{2}{\sqrt{\alpha}+1}\chi_0 + \frac{1}{(\sqrt{\alpha}+1)^2} = 0 \tag{237}$$

$$\implies \chi_0 = \frac{1}{\sqrt{\alpha}+1}. \tag{238}$$

Then, Equation (234), for the critical $\theta_c$ becomes:

$$\frac{\frac{\sqrt{\alpha}-1}{\sqrt{\alpha}+1}\alpha}{(\alpha-1)(1-\frac{1}{\sqrt{\alpha}+1})} = \frac{1}{\theta_c+1} \tag{239}$$

$$\implies \frac{\sqrt{\alpha}}{(\sqrt{\alpha}+1)} = \frac{1}{\theta_c+1} \tag{240}$$

$$\implies 1+\theta_c = 1+\frac{1}{\sqrt{\alpha}}, \tag{241}$$

so we conclude $\theta_c = \frac{1}{\sqrt{\alpha}}$. Then, for all $\alpha \geq \frac{1}{\sqrt{\alpha}}$, there will be an outlier in the spectrum.

