## A. Preliminaries

Recall the definition of $\hat{A}$:

$$\hat{A} := [\hat{a}_1, \ldots, \hat{a}_d] \in \mathbb{R}^{d \times d}, \tag{22}$$

where $\hat{a}_k$ solves Equation (1). We can first re-write as matrix optimization problem:

$$\hat{A} = \underset{\substack{A \in \mathbb{R}^{d \times d} \\ \text{diag}(A) = 0}}{\arg\min} \frac{1}{n} \|X - XA^\top\|_F^2 + \lambda \|A\|_F^2. \tag{23}$$

### A.1. Proof of 3.1

We solve this problem by imposing the KKT conditions in Equation (23). Let $g_l(A) = A_{l,l}$ and $F(A) = \frac{1}{n}\|X - XA^T\|_2^2 + \lambda\|A\|_F^2$. Then, the Lagrangian is given by:

$$\mathcal{L}(A) = F(A) - \sum_{l=1}^{d} \lambda_l g_l(A) = F(A) - \text{Tr}(\Lambda A) \tag{24}$$

$$= \frac{1}{n}\text{Tr}\left((I_d - A^T)X^T X(I_d - A)\right) + \lambda\text{Tr}(A^T A) - \text{Tr}(\Lambda A) \tag{25}$$

where $\Lambda = \text{diag}(\lambda_1, \ldots, \lambda_d)$. Then, differentiating with respect to $A$, we have:

$$\frac{\partial}{\partial A}\mathcal{L}(A) = -\frac{2}{n}X^T X + \frac{2}{n}X^T XA + 2\lambda A + \Lambda, \tag{26}$$

and imposing the stationarity condition of the Lagrangian we get:

$$\frac{\partial \mathcal{L}}{\partial A} = -\frac{2}{n}X^T X + \frac{2}{n}(X^T X + \lambda I_d)A + \Lambda = 0. \tag{27}$$

Denote $Q(\lambda) := (\hat{\Sigma} + \lambda I_d)^{-1}$. Then, solving for $A$ gives:

$$A = (\frac{1}{n}X^T X + \lambda I_d)^{-1}(\frac{1}{n}X^T X + \frac{1}{2}\Lambda) = Q(\lambda)(\hat{\Sigma} + \frac{1}{2}\Lambda). \tag{28}$$

In order to obtain $\Lambda$, we impose the restriction of the optimization problem:

$$\text{diag}(\hat{A}) = \text{diag}\left((\hat{\Sigma} + \lambda I_d))^{-1}(\hat{\Sigma} + \frac{1}{2}\Lambda)\right) = 0 \tag{29}$$

$$\iff \lambda_k Q(\lambda)_{k,k} = -2Q(\lambda)_{k,:}^T \hat{\Sigma}_{k,:} \tag{30}$$

$$\iff \lambda_k = \frac{-2}{Q(\lambda)_{k,k}}Q(\lambda)_{k,:}^T \hat{\Sigma}_{k,:}, \tag{31}$$

where we used the fact that $\Lambda$ is diagonal. Then, by defining $D(\lambda) \in \mathbb{R}^{d \times d}$ the diagonal matrix with entries:

$$D(\lambda)_{k,k} = \frac{\text{diag}(Q(\lambda)\hat{\Sigma})_{k,k}}{Q(\lambda)_{k,k}}, \tag{32}$$

we can write:

$$\hat{A} = Q(\lambda)(\hat{\Sigma} - D(\lambda)). \tag{33}$$

By adding and subtracting the $\frac{\lambda}{n}$ in (33), we also get:

$$\hat{A} = I_d - Q(\lambda)(D(\lambda) + \lambda I_d). \tag{34}$$

Note that

$$D(\lambda) + \lambda I_d = \frac{\operatorname{diag}(Q(\lambda)\hat{\Sigma})}{\operatorname{diag}(Q(\lambda))} + \lambda I_d \tag{35}$$

$$= \frac{1}{\operatorname{diag}(Q(\lambda))} \left( \operatorname{diag}(Q(\lambda)\hat{\Sigma}) + \lambda \operatorname{diag}(Q(\lambda)) \right) \tag{36}$$