# OpenReview forum: "A Random Matrix Theory of Masked Self-Supervised Learning"
_ICML.cc/2026/Conference — ICML 2026 regular_

### Official Review · Reviewer_h3CC · 2026-03-05

**Soundness:** 3
**Presentation:** 3
**Significance:** 3
**Originality:** 3
**Overall Recommendation:** 4
**Confidence:** 3

**Summary:**

This paper develops a random-matrix-theoretic understanding of masked self-supervised learning through a tractable linear proxy model. The authors formalize masked SSL as a “masked self-supervised ridge regression” procedure that trains a separate linear predictor for each masked coordinate and aggregates these predictors into a single matrix-valued estimator capturing cross-coordinate dependencies. They derive an explicit representation of this learned matrix in terms of the resolvent of the sample covariance with a nontrivial diagonal normalization, which enables precise high-dimensional analysis. Under a proportional asymptotic regime , the paper provides deterministic equivalents for key quantities, including the test reconstruction error and the spectral distribution of the learned matrix, yielding closed-form characterizations driven by the population covariance and the sample-to-dimension ratio. Finally, the theory is specialized to structured covariance models , illustrating how masked prediction behavior and performance depend on the underlying correlation structure and explaining regimes where masked self-supervised learning aligns with or differs from classical PCA-style reconstruction.

**Compliance With Llm Reviewing Policy:**

Affirmed.

**Key Questions For Authors:**

1. What is the most essential correspondence between your linear masked-SSR surrogate model and real masked pretraining methods (e.g., MAE/BERT-style training)?

2. How should the diagonal normalization term $\Lambda = [\mathrm{diag}(Q)]^{-1}$ in the aggregated matrix representation be interpreted statistically? Can you provide a more intuitive derivation or an equivalent formulation to clarify why this term is “inevitable” rather than merely a technical artifact?

3. Your main results rely on a specific proportional high-dimensional limit and distribution/moment assumptions. How robust are the conclusions when these assumptions are violated—for example, under non-Gaussianity, heavy tails, heteroscedasticity, or strongly local covariance structure that may not satisfy certain technical conditions? In such cases, do the deterministic-equivalent predictions for risk and spectral behavior remain approximately accurate?

**Limitations:**

yes

**Strengths And Weaknesses:**

Soundness: The work formalizes masked self-supervised learning by casting it as a constrained ridge regression problem solved separately for each coordinate and then aggregated into a matrix-valued predictor. Both the training objective and the generalization metric are explicitly defined, making the setup suitable for rigorous analysis and for direct comparison with empirical curves. A key technical contribution is an explicit representation of the learned matrix in terms of the resolvent of the sample covariance together with a diagonal normalization, which enables deterministic-equivalent tools from random matrix theory to yield precise high-dimensional limits. The analysis is conducted in a standard proportional asymptotic regime under linear Gaussian/sub-Gaussian assumptions with ridge regularization, and the theory is further instantiated on structured covariance models to support interpretability.

Presentation: The exposition follows a clear progression from the definition of masked self-supervised ridge regression, to a central structural lemma, to the main high-dimensional theorems on risk and spectral behavior, and finally to illustrative covariance case studies such as spiked models versus AR(1). The notation is consistent and aligns with common random-matrix conventions, which should be accessible to the intended theoretical/statistical learning audience. The contrast between global low-rank structure and local correlation structure (AR(1)) also helps build intuition about when masked prediction is likely to be beneficial.

Significance: Masked self-supervised learning is highly important in practice, yet rigorous high-dimensional statistical characterizations remain relatively limited. This paper provides an explicit and computable theoretical framework that could serve as a foundation for analyzing more complex masked SSL variants. The results deliver structural insights, notably that masked prediction tends to better exploit local or sequential correlation patterns and is not necessarily superior to PCA-style approaches designed to capture global low-rank factors, offering a useful lens on inductive bias across pretraining objectives. The explicit asymptotic expressions for risk and spectral quantities also make the framework a practical theoretical benchmark for comparing different regularization choices, masking mechanisms, and correlation structures.

Originality: The modeling perspective is distinctive in treating the outcome of masked SSL as a matrix-valued predictor and studying its spectral and generalization behavior as an integrated object rather than as isolated coordinate-wise regression tasks. Methodologically, incorporating the coupled “resolvent plus diagonal normalization” structure into a deterministic-equivalent random-matrix analysis represents a creative and valuable extension/combination of existing techniques. The use of phase-transition and universality viewpoints—such as covariance-independent spectral universality in specific regimes and BBP-type transitions in spiked models—provides additional novel angles on when learnable structure emerges in masked self-supervised learning.

---

> ### Author Rebuttal · Authors · 2026-03-30
>
> We thank the reviewer for their questions and feedback, which will definitely help improve the manuscript. We answer all the questions below.
>
> >*What is the most essential correspondence between your linear
> masked-SSR surrogate model and real masked pretraining methods
> (e.g., MAE/BERT-style training)*?
>
> Models like MAE/Bert rely on non-causal masked language modeling, whose goal is to predict a random masking of tokens in a sequence, with *no particular causal or directional structure*.  One can mathematically define this in the following way: Let $[\textsf{MASK}]$ be a special token and consider a masking map $C_{M}$ such that, given a subset of token indices $M\subset[L]$, replaces them with $\textsf{MASK}$. For example for $x= \textit{The capital of France is Paris}  (L=6)$ and $M=\{2,4\}$ and, we would get
> $C_{M}(x)=\textit{The [\textsf{MASK}] of [\textsf{MASK}] is Paris}$.
>
> The goal is then to recover $x$ from $C_{M}(x)$. In a general model, one has to model this via a masking distribution $q(\cdot| C_M(x))$. In the case where one models the language problem with $q$ being Gaussian, one recovers the SSR model described in our paper.
>
> > *How should the diagonal normalization term  $\Lambda =     [\mathrm{diag}(Q)]^{-1}$ in the aggregated matrix representation     be interpreted statistically? Can you provide a more intuitive  derivation or an equivalent formulation to clarify why this term     is “inevitable” rather than merely a technical artifact?*
>
> This term should be interpreted as a Lagrangian multiplier enforcing the constraint that a coordinate cannot be used to predict itself. This implies that the diagonal of the aggregated matrix predictor has to be $0$. This term is doing precisely that.
>
> >*Your main results rely on a specific proportional high-dimensional
> limit and distribution/moment assumptions. How robust are the
> conclusions when these assumptions are violated—for example, under non-Gaussianity, heavy tails, heteroscedasticity, or strongly local covariance structure that may not satisfy certain technical conditions? In such cases, do the deterministic-equivalent     predictions for risk and spectral behavior remain approximately accurate?*
>
> We would like to clarify first that our results do not require Gaussian covariates. Theorems 3.5 already allows for fairly general data of the form $x=\Sigma^{1/2} z$, where the entries of $z$ are independent, centered, have variance 1, and bounded $4+\varepsilon$ moments. Thus, the theory already covers a broad class of non-Gaussian distributions.
>
> Regarding heteroscedasticity or strongly local covariance structure, these are partly captured as long as the population covariance $\Sigma$ satisfies the assumptions of our theorems (in particular, bounded operator norm). In fact, the AR(1)/Toeplitz example is precisely meant to illustrate that the theory already applies to strongly local dependence structures, not only to low-rank or isotropic models.
>
> More broadly, we view the proportional asymptotic regime as the setting in which one can obtain exact and interpretable formulas. Extending these results beyond this regime, or to heavier-tailed models, is an interesting direction for future work, but would require substantial additional technical developments.

---

> > ### Author Rebuttal · Reviewer_h3CC · 2026-04-01
> >
> > My concerns have been adequately addressed. All questions are solved.

---

### Official Review · Reviewer_qRH2 · 2026-03-10

**Soundness:** 2
**Presentation:** 3
**Significance:** 3
**Originality:** 3
**Overall Recommendation:** 4
**Confidence:** 4

**Summary:**

This paper proposes an analysis of masked self-supervised learning (SSL) in the setting of linear regression, where the features are masked one by one, then predicted from all others through a regularized least-square optimization.  Levering tools from random matrix theory, this analysis provides asymptotic reconstruction error of SSL at comparably large numbers of features and samples, as well as the limiting spectrum of the matrix-valued predictor that aggregates the predictor vectors for different masked feature. Through two case studies that compare SSL with principal component analysis (PCA) in terms of reconstruction error, it is found that SSL is always surpassed by PCA when the correlations between features are captured by a one-dimensional spike, but it can perform better when the features exhibit a temporal relation described by an auto-regressive process of order 1. Experiments on synthetic data are reported to illustrate the validity of asymptotic results.

**Compliance With Llm Reviewing Policy:**

Affirmed.

**Final Justification:**

Overall it is decent work without any fatal weakness. However, there are statements on the technical difficulty that are potentially misleading.

**Key Questions For Authors:**

- It is not clear to me why the characterization of the matrix-valued predictor $\hat A$ "precludes a direct application of standard
random matrix results".  Since the diagonal elements of $Q(\lambda)$ concentrate around some deterministic limits, it suffices to find the deterministic equivalent $\bar Q(\lambda)$ of $Q(\lambda)$ to obtain the deterministic equivalent of $\hat A$ as $I_d-\bar Q(\lambda)[{\rm diag}(\bar Q(\lambda))]^{-1}$. Did I misunderstand?

- Can the expression of $L_{\rm App}$ in Equation (5) be simplified to $\frac{1}{d}{\rm Tr}([{\rm diag}(\Sigma)^{-1}]^{-1})$? If so, the discussion below Lemma 3.3 would be overly complicated.

- Is $L(PCA)_p$ obtained by replacing $\hat A$ in Equation (3) with $A_p^{\rm PCA}$ in Equation (15)? If so, wouldn't it be fairer to compare the performances of PCA and SSL at the same rank for $\hat A$ and $A_p^{\rm PCA}$?

**Limitations:**

There lacks a discussion on the potential extensions of the current analysis.

**Strengths And Weaknesses:**

Strengths

- Soundness: The problem setup is clearly laid out, the theoretical results are properly stated, discussed and verified through simulations.
- Presentation: The manuscript is well structured and easy to read.
- Significance: The proposed analysis helps advance the understanding of self-supervised learning, which is a key component in LLMs
- Originality: This work follows a long line of exact analyses of machine learning methods in the proportional regime, with a new focus on self-supervised learning.

Weaknesses

- Soundness: There might be some redundancies and ambiguities in the theoretical statements (See Key Questions for Authors).
- Presentation: The notations with lowercase $x$ and capital $X$ are confusing, for instance, $X_i$ and $x_i$ both stand for an observed data point. Since Section 4 is centered around the comparison between $L(SSR)$ and $L(PCA)_p$, the term $L(PCA)_p$ should be mathematically defined.
- Significance: The practical relevance of the case studies in Section 4 is not very clear (See Key Questions for Authors).
- Originality: I have some doubts on the technical novelty of the proposed analysis claimed by the authors (See Key Questions for Authors).

---

> ### Author Rebuttal · Authors · 2026-03-30
>
> We thank the reviewer for their questions and feedback, which will definitely help improve the manuscript. We answer all the questions below.
>
> >*The notations with lowercase  and capital  are confusing, for
> instance,  and  both stand for an observed data point. Since
> Section 4 is centered around the comparison between $L(SSR)$ and
> $L(PCA)$, the term  should be mathematically defined.*
>
> We thank the reviewer for noting these issues. We will follow this suggestion and revise the notation in the revised version of the paper.
>
> >*It is not clear to me why the characterization of the matrix-valued predictor  "precludes a direct application of standard random matrix results". Since the diagonal elements of  $Q(\lambda)$ concentrate around some deterministic limits, it suffices to find the deterministic equivalent $\bar{Q}(\lambda)$ of $Q(\lambda)$ to obtain the deterministic equivalent of $\hat{A}$ as $I_d - \bar{Q}[\mathrm{diag}(\bar{Q}(\lambda))]^{-1}$. Did I misunderstand?*
>
> The reason this falls outside the classic results of random matrix theory comes from the fact that the matrix $\mathrm{diag}(Q(\lambda))$ and $Q(\lambda)$ are not independent. The reviewer is correct in their observation about the diagonal matrix being concentrated, but after this is established, the deterministic equivalent of the product $Q(\lambda) [\mathrm{diag}(Q(\lambda))]^{-1}$ is not the product of the deterministic equivalents, as the presence of a diagonal matrix fundamentally changes the spectrum of the whole matrix. The main technical component of Theorem 3.8 is deriving the deterministic equivalent of this product.
>
> >*Can the expression of $L_\mathrm{App}$ in Equation (5) be
> simplified (..)? If so, the discussion below Lemma 3.3 would be
> overly complicated.*
>
> We thank the reviewer for pointing this out. Indeed this is the case. We noticed this fact after submission and will simplify this expression in the revised version of the manuscript.
>
> >Is $L(PCA)\_p$ obtained by replacing $\hat{A}$ in Equation (3) with  $\hat{A}\_{PCA}$ in Equation (15)? If so, wouldn't it be fairer to  compare the performances of PCA and SSL at the same rank for  $\hat{A}$ and  $\hat{A}\_{PCA}$ ?
>
> Yes, $L(PCA)\_p$ is computed by evaluating Eq. (3) with $\hat{A}\_{PCA}$ from Eq. (15). Regarding the comparison at fixed rank, we would like to emphasize that the two methods operate under fundamentally different constraints. The SSR estimator $\hat{A}$ is (almost surely) full-rank and is not designed for dimensionality reduction, whereas PCA explicitly enforces a rank-$p$ constraint.
>
> Our results show that, in order to achieve performance comparable to SSR, PCA must take $p = \Theta(d)$, i.e., retain a number of components proportional to the ambient dimension. In this regime, PCA effectively ceases to perform dimensionality reduction and instead uses nearly all directions. Therefore, we think that comparing at fixed (low) rank highlights an inherent limitation of PCA in this setting, rather than an artifact of the comparison.

---

> > ### Author Rebuttal · Reviewer_qRH2 · 2026-04-04
> >
> > I thank the authors for their reply. Regarding the question on the technical difficulty, did the authors mean to say that the deterministic equivalent of $Q(\lambda)[{\rm diag}( Q(\lambda))]^{-1}$ is not given by $\bar Q(\lambda)[{\rm diag}(\bar Q(\lambda))]^{-1}$ with $\bar Q(\lambda)$ being the deterministic equivalent of $Q(\lambda)$ as understood in Definition 4 of Couillet & Liao (2022)? If so, could the authors provide more details about how the deterministic equivalent of $Q(\lambda)[{\rm diag}( Q(\lambda))]^{-1}$ differs from $\bar Q(\lambda)[{\rm diag}(\bar Q(\lambda))]^{-1}$?
> >
> > =========================================================
> > Comments after Reply Rebuttal Comment by Authors
> >
> > Thanks for the reply. As the $[{\rm diag}( Q(\lambda))]^{-1}$ is well approximated by $[{\rm diag}( \bar Q(\lambda))]^{-1}$ with small error in spectral norm, it behaves similarly to a deterministic matrix. I still don't see why it is needed to handle the dependence between $[{\rm diag}( Q(\lambda))]^{-1}$ and $Q(\lambda)$, even for the resolvent of $\hat A$

---

> > > ### Author Response · Authors · 2026-04-04
> > >
> > > Let $\hat A = Q(\lambda) [diag(Q(\lambda))]^{-1}$. The reviewer is absolutely correct that a deterministic equivalent of $\hat A$ (understood as the entrywise expectation of $\hat A$ up to a correction term with vanishing operator norm) is obtained by taking the standard deterministic equivalent of $Q(\lambda)$ and multiplying it by $[diag(\bar Q(\lambda))]^{-1}$.
> > >
> > > However, this is not the object we intended to describe. While $\hat A$ is generally non-symmetric, its eigenvalues are real because it is similar to a symmetric matrix, which we denote here by $\tilde A$ (see Remark 3.2 for details). Our goal is to characterize the deterministic equivalent of the **resolvent** of this symmetric matrix $\tilde A$ at a spectral parameter z in the upper complex plane. This characterization will then further allow us to study the limiting spectrum of the aggregate self-supervised estimator $\hat A$.
> > >
> > > Conceptually, $Q(\lambda)$ is itself a resolvent. In contrast, we study the resolvent of a diagonally normalized version of this resolvent, which introduces additional structure and technical challenges. This characterization is given in Theorem 3.8.
> > >
> > > We thank the reviewer for pointing out this terminological ambiguity. To avoid confusion, we will revise the manuscript accordingly: instead of referring to the deterministic equivalent of $\hat A$, we will explicitly refer to that of the resolvent of $\tilde A$.

---

### Official Review · Reviewer_YEUn · 2026-03-12

**Soundness:** 3
**Presentation:** 3
**Significance:** 3
**Originality:** 3
**Overall Recommendation:** 5
**Confidence:** 3

**Summary:**

In this paper, the authors consider the masked self-supervised learning problem, where the goal is to predict the masked coordinate. For a given data matrix, an aggregate predictor matrix $\hat A$ (called the SSR matrix) is constructed, and asymptotic limit of the risk is proved. The proof is based on the analysis of the resolvent as in random matrix theory, where the main idea is that the resolvent of a random matrix can be well-approximated by a deterministic counterpart. The authors also presented a BBP-type phase transition for the top eigenvalues of the SSR matrix.

**Compliance With Llm Reviewing Policy:**

Affirmed.

**Final Justification:**

The authors answered my question adequately. I will maintain my score.

**Key Questions For Authors:**

I wonder if the generalization error in Theorem 3.5 and the training error in Theorem 3.7 exhibit the phase transition similar to the BBP transition, since the solution to the self-consistent equation involving the resolvent may heavily depend on the outlier eigenvalues.

**Limitations:**

Yes

**Strengths And Weaknesses:**

Strengths
- The results establish a robust mathematical framework for the analysis of the generalization error and training error with simple and clear formulas.
- The theoretical analysis based on random matrix theory is profound and rigorous.
- The manuscript is well-written and the results are well-presented.

Weaknesses
- The model is rather too simplified, e.g., linear SSL or spiked covariance model.

---

> ### Author Rebuttal · Authors · 2026-03-30
>
> We thank the reviewer for their positive assessment of our work and for their useful feedback. We answer their comments and questions below.
>
> >The model is rather too simplified, e.g., linear SSL or spiked covariance model.
>
> We agree that the model we study is rather stylized. Our goal is to obtain a precise and tractable understanding of masked SSR, and to the best of our knowledge, this is the first work providing an exact asymptotic analysis in the fully masked setting. As is common in theoretical work, we view this as a first step toward more realistic scenarios, including multi-mask objectives, higher-dimensional embeddings, non-linear architectures, and more general loss functions. At the same time, we emphasize that even this simplified setting already requires new technical developments beyond standard RMT, due to the strong dependencies induced by the matrix-valued predictor.
>
> >*I wonder if the generalization error in Theorem 3.5 and the     training error in Theorem 3.7 exhibit the phase transition similar     to the BBP transition, since the solution to the self-consistent     equation involving the resolvent may heavily depend on the outlier eigenvalues.*
>
> We do not expect a BBP transition in the training or generalization error. The reason for this is that in the model we propose, the perturbation adds an eigenvector to the covariance matrix with coordinates of order $\sim \frac{1}{\sqrt{d}}$. However, in order to have a low reconstruction error (that is, low generalization/training error), it is necessary to recover other eigenvectors as well, but this spike does not add strong correlations with the rest of the coordinates (note that the resulting covariance has diagonal entries of order $1 + \frac{\theta}{d}$, and off-diagonal entries of order $\frac{\theta}{d}$).

---

> > ### Author Rebuttal · Reviewer_YEUn · 2026-04-01
> >
> > The authors answered my question adequately.

---

### Official Review · Reviewer_pnqo · 2026-03-13

**Soundness:** 3
**Presentation:** 4
**Significance:** 3
**Originality:** 4
**Overall Recommendation:** 5
**Confidence:** 3

**Summary:**

This paper provides a rigorous high-dimensional analysis of Masked Self-Supervised Learning (SSL) through the lens of linear ridge regression (SSR). The core contribution is the development of a Random Matrix Theory (RMT) framework to characterize a joint, matrix-valued predictor—an object that arises from aggregating predictions across multiple masking patterns. The authors derive sharp asymptotic limits for the pre-training risk and the spectral density of this predictor. These theoretical tools are then applied to two case studies (Spiked Covariance and AR(1) models) to illustrate the regimes where SSR either fails or succeeds relative to PCA.

**Compliance With Llm Reviewing Policy:**

Affirmed.

**Final Justification:**

I thank the authors for their clear and targeted rebuttal. The rebuttal has addressed my questions. I maintain my recommendation of Accept.

**Key Questions For Authors:**

See weaknesses.

**Limitations:**

yes

**Strengths And Weaknesses:**

Strengths:
1. Novel Theoretical Angle: Most SSL theory focuses on contrastive learning; this work fills a significant gap by providing a precise mathematical treatment of masked reconstruction.
2. Analytical Depth: The derivation of the BBP-type phase transition for the aggregate predictor is a sophisticated application of RMT that reveals when SSL begins to "see" latent signals.
3. Practical Implications: The finding that masked modeling is specifically optimized for localized sequential structures (like AR(1) processes) provides a formal justification for its success in NLP and sequence modeling over global methods like PCA.

Weaknesses:
1. Multi-Masking: In practice, models like BERT or MAE mask multiple coordinates randomly and simultaneously, rather than aggregating individual "leave-one-out" predictions. Is there an extension of the SSR framework to random multi-mask objectives, and would the same spectral / risk picture still hold?
2. Intuition for Sequential Advantage: While the paper proves that SSR can outperform PCA in AR(1) models, it fails to provide a compelling intuitive explanation for why this occurs. The result remains largely an asymptotic observation based on "degrees of freedom" calculations. The paper would be much stronger if it could explain the underlying mechanism—perhaps in terms of how the masking objective acts as a localized filter that matches the exponentially decaying structure of AR(1) data.

---

> ### Author Rebuttal · Authors · 2026-03-30
>
> We thank the reviewer for their positive assessment of our work and for their useful feedback. We answer all questions and comments below.
>
>
> >*Multi-Masking: In practice, models like BERT or MAE mask multiple     coordinates randomly and simultaneously, rather than aggregating     individual "leave-one-out" predictions. Is there an extension of     the SSR framework to random multi-mask objectives, and would the     same spectral / risk picture still hold?*
>
> This is a very interesting question, which we are currently looking at. In our problem, the fact that the mask is always of size $1$ implies a restriction on the diagonal of the matrix being $0$, and therefore the Lagrange multiplier that appears in the solution to the optimization problem in Eq. (32) is a diagonal matrix. For multiple masking, this constraint is more complicated and requires a significantly more involved analysis.
>
> >*Intuition for Sequential Advantage: While the paper proves that SSR
> can outperform PCA in AR(1) models, it fails to provide a compelling
> intuitive explanation for why this occurs. The result remains largely
> an asymptotic observation based on ``degrees of freedom'' calculations.
> The paper would be much stronger if it could explain the underlying
> mechanism—perhaps in terms of how the masking objective acts as a
> localized filter that matches the exponentially decaying structure of
> AR(1) data.*
>
> We thank the reviewer for this insightful suggestion and agree that clarifying the mechanism is important.
>
> The key point is a mismatch between local vs. global structure. In an AR(1) model, correlations decay with distance, so each coordinate is mainly predictable from its neighbors. The masked objective naturally exploits this: predicting one coordinate from the rest effectively focuses on nearby coordinates, acting like a local filter. In contrast, PCA is driven by global eigenvectors, which in this setting are delocalized (spread across all coordinates). As a result, PCA captures global variation but does not align well with the local predictive structure needed for reconstruction.
>
> This is reflected in Theorem 3.5: the SSR error depends on how eigenvectors interact with a diagonal reweighting that encodes coordinate-wise predictability. When eigenvectors are delocalized, this favors SSR. In the revised manuscript, we will highlight this intuition more clearly by elaborating on this distinction and reorganizing the discussion around Theorem 3.5.

---

> > ### Author Rebuttal · Reviewer_pnqo · 2026-04-04
> >
> > I thank the authors for their clear and targeted rebuttal. The rebuttal has addressed my questions. I maintain my recommendation of Accept.

---

### Decision · Program_Chairs · 2026-04-30

**Decision:**

Accept (regular)

**Comment:**

Masked self-supervised learning is widely used in LLMs, and this paper makes a very good theoretical contribution towards understanding it in a stylized setting. The reviewers agree on the technical strengths of the paper, including modeling the output of SSL as an aggregrated matrix-valued predictor, and obtaining precise formulas for the generalization error and its spectral structure in the proportional asymptotics regime. The paper is well-written and the analysis is rigorous. I would suggest including a discussion of limitations of the current model and the directions for future work (that the authors mentioned in the rebuttal).